# Fair Rank Aggregation

**Diptarka Chakraborty**
School of Computing
National University of Singapore
diptarka@comp.nus.edu.sg

**Syamantak Das**
Department of Computer Science and Engineering
Indraprastha Institute of Information Technology, Delhi
syamantak@iiit.ac.in

**Arindam Khan**
Department of Computer Science and Automation
Indian Institute of Science, Bengaluru
arindamkhan@iisc.ac.in

**Aditya Subramanian**
Department of Computer Science and Automation
Indian Institute of Science, Bengaluru
adityasubram@iisc.ac.in

## Abstract

Ranking algorithms find extensive usage in diverse areas such as web search, employment, college admission, voting, etc. The related rank aggregation problem deals with combining multiple rankings into a single aggregate ranking. However, algorithms for both these problems might be biased against some individuals or groups due to implicit prejudice or marginalization in the historical data. We study ranking and rank aggregation problems from a fairness or diversity perspective, where the candidates (to be ranked) may belong to different groups and each group should have a fair representation in the final ranking. We allow the designer to set the parameters that define fair representation. These parameters specify the allowed range of the number of candidates from a particular group in the top-$k$ positions of the ranking. Given any ranking, we provide a fast and exact algorithm for finding the closest fair ranking for the Kendall tau metric under *strong fairness*, i.e., when the final ranking is fair for all values of $k$. We also provide an exact algorithm for finding the closest fair ranking for the Ulam metric under strong fairness when there are only $\mathcal{O}(1)$ number of groups. Our algorithms are simple, fast, and might be extendable to other relevant metrics. We also give a novel meta-algorithm for the general rank aggregation problem under the fairness framework. Surprisingly, this meta-algorithm works for any generalized mean objective (including center and median problems) and any fairness criteria. As a byproduct, we obtain 3-approximation algorithms for both center and median problems, under both Kendall tau and Ulam metrics. Furthermore, using sophisticated techniques we obtain a $(3 - \varepsilon)$-approximation algorithm, for a constant $\varepsilon > 0$, for the Ulam metric under strong fairness.

## 1 Introduction

Ranking a set of candidates or items is a ubiquitous problem arising in diverse areas ranging from social choice theory [BCE+16] to information retrieval [Har92]. Given a set of $d$ candidates and a set of $n$ different, potentially conflicting, rankings of these candidates, one fundamental task is to

36th Conference on Neural Information Processing Systems (NeurIPS 2022).

determine a single ranking that best summarizes the preference orders in the individual rankings. This summarizing task, popularly termed *rank aggregation*, has been widely studied from a computational viewpoint over the last two decades [DKNS01, FKS03, GL11, ASCPX13]. Most well-studied rank aggregation paradigms are *median rank aggregation* (or simply *rank aggregation*) [Kem59, You88, YL78, DKNS01] and *maximum rank aggregation* [BBGH15, BBD09, Pop07], which are based on finding the *median* and *center* of the given set of rankings, respectively.

Recently, fairness and diversity have become a natural prerequisite for ranking algorithms where individuals are rated for access to goods and services or ranked for seeking facilities in education (e.g., obtaining scholarship or admission), employment (e.g., hiring or promotion in a job), medical (e.g., triage during a pandemic), or economic opportunities (e.g., loan lending). Some concrete examples include university admissions through affirmative action in the USA [Des05] or the reservation system in jobs in India [Bor10], where we want rankings to be fair to mitigate the prevalent disparities due to historical marginalization. Rankings not being fair may risk promoting extreme ideology [CHRG16] or certain stereotypes about dominating/marginalized communities based on sensitive attributes like gender or race [KMM15, BCZ$^+$16]. There has been a series of works on fair ranking algorithms, see [ZBC$^+$17, AJSD19, Cas19, GSB21, GDL21, PPM$^+$22, ZYS21] and the references therein.

A substantial literature on algorithmic fairness focuses on *group fairness* to facilitate *demographic parity* [DHP$^+$12] or *equal opportunity* [HPS16]: typically this is done by imposing fairness constraints which require that top-$k$ positions in the ranking contain *enough* candidates from *protected* groups that are typically underrepresented due to prevalent discrimination (e.g., due to gender, caste, age, race, sex, etc.). In many countries, group fairness constraints are being enforced by legal norms [Eur, USD]. For example, in Spain 40% of candidates for elections in large voting districts must be women [Ver10], in India 10% of the total recruitment for civil posts and services in government are reserved for people from Economically Weaker Society (EWS) [Sin19], etc.

In this paper, we study group fairness, more specifically proportional fairness (sometimes also referred to as $p$-fairness [BCPV96]). Inspired by the *Disparate Impact* doctrine, this notion of fairness mandates that the output of an algorithm must contain a fair representation of each of the 'protected classes' in the population. In the context of ranking, the set of candidates is considered to be partitioned into $g$ groups $G_1, G_2, \ldots, G_g$. For each group $G_i, i \in [g]$, we have two parameters $\alpha_i \in [0,1], \beta_i \in [0,1]$. A ranking $\pi$ of the set of items is called *proportionally fair* if for every position $k \in \{1, 2, \ldots, n\}$ and for every group $G_i$, the following two properties are satisfied: (a) *Minority Protection*: The number of items from group $G_i$, which are in the top-$k$ positions $\pi(1), \pi(2), \ldots, \pi(k)$, is at least $\lfloor \alpha_i \cdot k \rfloor$, and (b) *Restricted Dominance*: The number of items from group $G_i$, which are in the top-$k$ positions $\pi(1), \pi(2), \ldots, \pi(k)$, is at most $\lceil \beta_i \cdot k \rceil$.

To compare different rankings several distance functions have been considered defined on the set of permutations/rankings, such as Kendall tau distance [Ken38, DG77, Kem59, You88, YL78, DKNS01, ACN08, KMS07, KV10] (also called *Kemeny distance* in case of rank aggregation), Ulam distance [AD99, CMS01, CK06, AK10, AN10, NSS17, BS19, CDK21, CGJ21], Spearman footrule distance [Spe04, Spe06, DG77, DKNS01, KV10, BBGH15], etc. Among these, Kendall tau distance is perhaps the most common measure used in ranking as it is the only known measure to simultaneously satisfy several required properties such as neutrality, consistency, and the extended Condorcet property [Kem59, You88]. The Ulam metric is another widely-used measure in practice as it is also a simpler variant of the general edit distance metric which finds numerous applications in computational biology, DNA storage system, speech recognition, classification, etc. (e.g., see [CMS01, CDK21, CGJ21]).

One natural computational question related to fairness in ranking is, given a ranking, how to find its closest fair ranking under $p$-fairness. Celis et al. [CSV18] considered this problem and gave exact and approximation algorithms under several ranking metrics such as discounted cumulative gain (DCG), Spearman footrule, and Bradley-Terry. However, their algorithms do not extend to Kendall tau and Ulam metric, two of the most commonly used ranking metrics.

Fair rank aggregation is relatively less studied. Recently, Kulman et al. [KR20] initiated the study of fair rank aggregation under Kendall tau metric. However, their fairness notion is based on *top-$k$ statistical parity* and *pairwise statistical parity*. These notions are quite restricted. For example, their results only hold for binary protected attributes (i.e., $g = 2$) and and does not satisfy $p$-fairness. Informally, pairwise statistical parity considers pairs of items from different groups in an aggregated manner and does not take into account the actual rank of the items in the final ranking. See [WISR22]

for an example on why the fairness notion in [KR20] does not satisfy $p$-fairness. In fact, as $p$-fairness satisfies statistical parity for all top $k$-positions in the ranking, it is a much stronger notion compared to statistical parity. Thus achieving $p$-fairness is a significantly more challenging problem.

## 1.1 Our Contributions.

Our first main contribution is an exact algorithm for the closest fair ranking (CFR) problem under proportional fairness (see Definition 2.4) for Kendall tau and Ulam metrics. For the Kendall tau metric, we give the *first exact algorithm* for the closest fair ranking problem (Theorem 3.4). Our algorithm is simple and based on a greedy strategy; however, the analysis is delicate. It exploits the following interesting and perhaps surprising fact. Under the Kendall tau metric, given a fixed (possibly unfair) ranking $\pi$, there exists a closest fair ranking $\pi'$ to $\pi$ such that for every group $G_i, i \in [g]$, the relative ordering of elements in $G_i$ remains unaltered in $\pi'$ compared to $\pi$ (Claim 3.3). Then, for the *Ulam metric*, we give a polynomial time dynamic programming algorithm for the closest fair ranking problem when the number of groups $g$ is a constant (Theorem 3.10). In practice, the number of protected classes is relatively few, and hence our result gives an efficient algorithm for such cases.

Our second significant contribution is the study of *rank aggregation problem under a generalized notion of proportional fairness*. One of our main contributions is to develop a novel algorithmic framework for the fair rank aggregation that solves a wide variety of rank aggregation objectives satisfying such generic fairness constraints. An essential takeaway of our work is that a set of potentially biased rankings can be aggregated into a fair ranking with only a small loss in the 'quality' of the ranking. We study $q$-mean Fair Rank Aggregation (FRA), where given a set of rankings $\pi_1, \pi_2, \ldots, \pi_n$, a (dis)similarity measure (or distance function) $\rho$ between two rankings, and any $q \geq 1$, the task is to determine a *fair* ranking $\sigma$ that minimizes the generalized mean objective: $\left(\sum_{i=1}^{n} \rho(\pi_i, \sigma)^q\right)^{1/q}$. We would like to emphasize that in general, $q$-mean objective captures two classical data aggregation tasks: One is *median* which asks to minimize the sum of distances (i.e., $q = 1$) and another is *center* which asks to minimize the maximum distance to the input points (i.e., $q = \infty$).

We show generic reductions of the $q$-mean Fair Rank Aggregation (FRA) to the problem of determining the *closest fair ranking* (CFR) to a given ranking. More specifically, we show that any $c$-approximation algorithm for the closest fair ranking problem can be utilized as a blackbox to give a $(c + 2)$-approximation to the FRA for any $q \geq 1$ (Theorem 4.3). This result is oblivious to the specifics of the (dis)similarity measure and only requires the measure to be a metric. Using the exact algorithms for the CFR for the Kendall tau, Spearman footrule, and Ulam metrics (for constantly many groups), we thus obtain 3-approximation algorithms for the FRA problem under these three (dis)similarity measures, respectively. Further, we provide yet another simple algorithm that even breaks below 3-factor for the Ulam metric. For $q = 1$, by combining the above-stated 3-approximation algorithm with an additional procedure, we achieve a $(3 - \varepsilon)$-approximation factor (for some $\varepsilon > 0$) for the FRA under the Ulam, for constantly many groups (Theorem 4.11). We also provide another reduction from FRA to one rank aggregation computation (without fairness) and a CFR computation (Theorem 4.8), and as a corollary get an $\mathcal{O}(d^3 \log d + n^2 d)$-time algorithm for Spearman footrule when $q = 1$ (Corollary 4.10). We summarize our main results in Table 1.1.

| Problem | Metric | #Groups | Approx Ratio | Runtime | Reference |
|---|---|---|---|---|---|
| **CFR** | Kendall tau | Arbitrary | Exact | $\mathcal{O}(d^2)$ | Theorem 3.4 |
| | Ulam | Constant | Exact | $\mathcal{O}(d^{g+2})$ | Theorem 3.10 |
| | Spearman footrule | Arbitrary | Exact | $\mathcal{O}(d^3 \log d)$ | [CSV18] |
| **FRA** | Kendall tau | Arbitrary | 3 | $\mathcal{O}(nd^2 + n^2 d \log d)$ | Corollary 4.5 |
| | Ulam | Constant | 3 | $\mathcal{O}(nd^{g+2} + n^2 d \log d)$ | Corollary 4.7 |
| | Ulam ($q = 1$) | Constant | $3 - \varepsilon$ | $\text{poly}(n, d)$ | Theorem 4.11 |
| | Spearman footrule | Arbitrary | 3 | $\mathcal{O}(d^3 \log d + n^2 d + nd^2)$ | Corollary 4.10 |

**Comparison with concurrent work.** Independently and concurrently to our work, Wei et al. [WISR22] considers the fair ranking problem under a setting that is closely related to ours. However, the fairness criteria in their work are much more restrictive compared to ours as follows.

Their algorithms for CFR are only designed for a special case of our formulation where for each group $G_i$ and any position $k$ in the output ranking, $\alpha_i = \beta_i = p(i)$, where $p(i)$ denotes the proportion of group $G_i$ in the entire population. Further, under the Kendall tau metric, they give a polynomial time exact algorithm for CFR only for the special case of binary groups ($g = 2$). They also give additional algorithms for multiple groups - an exact algorithm that works in time exponential in the number of groups and a polynomial time 2-approximation. In contrast, we fully resolve the CFR problem under the Kendall tau metric by giving a simple polynomial time algorithm for the case of multiple groups and any arbitrary bounds on $\alpha_i$ and $\beta_i$ for each group $G_i$. Further, we give the first results for CFR and FRA under the Ulam metric as well.

## 1.2 Other related work

Recent years have witnessed a growing concern over ML algorithms or, more broadly, automated decision-making processes being biased [BHN17, MMS+21]. This bias or unfairness might stem both from implicit bias in the historical dataset or from prejudices of human agents who are responsible for generating part of the input. Thus fair algorithms have received recent attention in machine learning and related communities. In particular, the notion of group fairness have been studied in classification [HV19], clustering [CKLV17], correlation clustering [AEKM20], resource allocation [PKL21], online learning [PGNN20], matroids and matchings [CKLV19]. Specially, fair clustering problem is closely related to our problem. Fair rank aggregation can be considered as the fair 1-clustering problem where the input set is a set of rankings. See [HJV19, CFLM19, BCFN19, BIO+19] for more related work on fair clustering.

## 2 Preliminaries

**Notations.** For any $n \in \mathbb{N}$, let $[n]$ denote the set $\{1, 2, \ldots, n\}$. We refer to the set of all permutations/rankings over $[d]$ by $\mathcal{S}_d$. Throughout this paper we consider any permutation $\pi \in \mathcal{S}_d$ as a sequence of numbers $a_1, a_2, \ldots, a_d$ such that $\pi(i) = a_i$, and we say that the *rank* of $a_i$ is $i$. For any two $x, y \in [d]$ and a permutation $\pi \in \mathcal{S}_d$, we use the notation $x <_\pi y$ to denote that the rank of $x$ is less than that of $y$ in $\pi$. For any subset $I = \{i_1 < i_2 < \cdots < i_r\} \subseteq [d]$, let $\pi(I)$ be the sequence $\pi(i_1), \pi(i_2), \ldots, \pi(i_r)$ (which is essentially a subsequence of the sequence represented by $\pi$). When clear from the context, we use $\pi(I)$ also to denote the set of elements in the sequence $\pi(i_1), \pi(i_2), \ldots, \pi(i_r)$. For any $k \in [d]$ and a permutation $\pi \in \mathcal{S}_d$, we refer to $\pi([k])$ as the $k$-length *prefix* of $\pi$. For any prefix $P$, let $|P|$ denote the length of that prefix. For any two prefixes $P_1, P_2$ of a given string, we use $P_1 \subseteq P_2$ to denote $|P_1| \leq |P_2|$.

**Distance measures on rankings.** There are different distance functions being considered to measure the dissimilarity between any two rankings/permutations. Among them, perhaps the most commonly used one is the *Kendall tau distance*.

**Definition 2.1** (Kendall tau distance)**.** Given two permutations $\pi_1, \pi_2 \in \mathcal{S}_d$, the *Kendall tau distance* between them, denoted by $\mathcal{K}(\pi_1, \pi_2)$, is the number of pairwise disagreements between $\pi_1$ and $\pi_2$, i.e.,
$$\mathcal{K}(\pi_1, \pi_2) := |\{(a, b) \in [d] \times [d] \mid a <_{\pi_1} b \text{ but } b <_{\pi_2} a\}|.$$

Another important distance measure is the *Spearman footrule* (aka *Spearman's rho*) which is essentially the $\ell_1$-norm between two permutations.

**Definition 2.2** (Spearman footrule distance)**.** Given two permutations $\pi_1, \pi_2 \in \mathcal{S}_d$, the *Spearman footrule distance* between them is defined as $\mathcal{F}(\pi_1, \pi_2) := \sum_{i \in [d]} |\pi_1(i) - \pi_2(i)|$.

Another interesting distance measure is the *Ulam distance* which counts the minimum number of character move operations between two permutations [AD99]. This definition is motivated by the classical *edit distance* that is used to measure the dissimilarity between two strings. A character move operation in a permutation can be thought of as "picking up" a character from its position and then "inserting" that character into a different position[1].

---

[1] One may also consider one deletion and one insertion operation instead of a character move, and define the Ulam distance accordingly as in [CMS01].

**Definition 2.3** (Ulam distance). Given two permutations $\pi_1, \pi_2 \in \mathcal{S}_d$, the *Ulam distance* between them, denoted by $\mathcal{U}(\pi_1, \pi_2)$, is the minimum number of character move operations that is needed to transform $\pi_1$ into $\pi_2$.

Alternately, the Ulam distance between $\pi_1, \pi_2$ can be defined as $d - |\mathsf{LCS}(\pi_1, \pi_2)|$, where $|\mathsf{LCS}(\pi_1, \pi_2)|$ denotes the length of a *longest common subsequence* between the sequences $\pi_1$ and $\pi_2$.

**Fair rankings.** We are given a set $C$ of $d$ candidates, which are partitioned into $g$ groups. We call a ranking (of these $d$ candidates) *fair* if all sufficiently large prefixes of it have certain proportion of representatives from each group. Formally,

**Definition 2.4** (($\bar{\alpha}, \bar{\beta}$)-$k$-fair ranking). Consider a set $C$ of $d$ candidates partitioned into $g$ groups $G_1, \ldots, G_g$, and $\bar{\alpha} = (\alpha_1, \ldots, \alpha_g) \in [0,1]^g, \bar{\beta} = (\beta_1, \ldots, \beta_g) \in [0,1]^g, k \in [d]$. A ranking $\pi \in \mathcal{S}_d$ is said to be ($\bar{\alpha}, \bar{\beta}$)-$k$-*fair* if for any prefix $P$ of length at least $k$, of $\pi$ and each group $i \in [g]$, there are at least $\lfloor \alpha_i \cdot |P| \rfloor$ and at most $\lceil \beta_i \cdot |P| \rceil$ elements from the group $G_i$ in $P$, i.e.,

$$\forall_{\text{prefix } P : |P| \geq k}, \ \forall_{i \in [g]}, \ \lfloor \alpha_i \cdot |P| \rfloor \leq |P \cap G_i| \leq \lceil \beta_i \cdot |P| \rceil.$$

We also define a weak fairness notion that preserves the proportionate representation only for a fixed $k$-length prefix.

**Definition 2.5** (($\bar{\alpha}, \bar{\beta}$)-weak $k$-fair ranking). Consider a set $C$ of $d$ candidates partitioned into $g$ groups $G_1, \ldots, G_g$, and $\bar{\alpha} = (\alpha_1, \ldots, \alpha_g) \in [0,1]^g, \bar{\beta} = (\beta_1, \ldots, \beta_g) \in [0,1]^g, k \in [d]$. A ranking $\pi \in \mathcal{S}_d$ is said to be ($\bar{\alpha}, \bar{\beta}$)-*weak $k$-fair* if for the $k$-length prefix $P$ of $\pi$ and each group $i \in [g]$, there are at least $\lfloor \alpha_i \cdot k \rfloor$ and at most $\lceil \beta_i \cdot k \rceil$ elements from the group $G_i$ in $P$, i.e.,

$$\forall_{i \in [g]}, \ \lfloor \alpha_i \cdot k \rfloor \leq |P \cap G_i| \leq \lceil \beta_i \cdot k \rceil.$$

Note, an ($\bar{\alpha}, \bar{\beta}$)-$k$-fair ranking is also ($\bar{\alpha}, \bar{\beta}$)-weak $k$-fair, but the converse need not be true. We would like to emphasize that all the results presented in this paper hold for both ($\bar{\alpha}, \bar{\beta}$)-$k$-fairness and ($\bar{\alpha}, \bar{\beta}$)-weak $k$-fairness.

## 3 Closest Fair Ranking

In this section, we consider the problem of computing the closest fair ranking of a given input ranking. Below we formally define the problem.

**Definition 3.1** (Closest fair ranking problem). Consider a metric space $(\mathcal{S}_d, \rho)$ for a $d \in \mathbb{N}$. Given a ranking $\pi \in \mathcal{S}_d$ and $\bar{\alpha}, \bar{\beta} \in [0,1]^g$ for some $g \in \mathbb{N}, k \in [d]$, the objective of the *closest fair ranking problem* (resp. closest weak fair ranking problem) is to find a ($\bar{\alpha}, \bar{\beta}$)-$k$-fair ranking (resp. ($\bar{\alpha}, \bar{\beta}$)-weak $k$-fair ranking) $\pi^* \in \mathcal{S}_d$ that minimizes the distance $\rho(\pi, \pi^*)$.

Unless stated explicitly, we consider the notion of ($\bar{\alpha}, \bar{\beta}$)-$k$-fairness (as opposed to the weak notion) in all the results presented in this section. The following algorithm assumes that a fair ranking exists, and finds one in this case. If such a ranking does not exist, then the procedure might return an arbitrary ranking. However, it is possible to check in linear time whether a given ranking is fair or not. Hence, we can correctly output a solution if one exists, and return 'no solution' otherwise.

### 3.1 Closest fair ranking under Kendall tau metric

**Closest weak fair ranking.** We first show that we can compute a closest weak fair ranking under the Kendall tau metric exactly in linear time.

**Theorem 3.2.** *There exists a linear time algorithm that, given a ranking $\pi \in \mathcal{S}_d$, a partition of $[d]$ into $g$ groups $G_1, \ldots, G_g$ for some $g \in \mathbb{N}$, and $\bar{\alpha} = (\alpha_1, \ldots, \alpha_g) \in [0,1]^g, \bar{\beta} = (\beta_1, \ldots, \beta_g) \in [0,1]^g, k \in [d]$, outputs a closest ($\bar{\alpha}, \bar{\beta}$)-weak $k$-fair ranking under the Kendall tau distance.*

Let us first describe the algorithm. Our algorithm follows a simple greedy strategy. For each group $G_i$, it picks the top $\lfloor \alpha_i k \rfloor$ elements according to the input ranking $\pi$, and add them in a set $P$. If $P$ contains $k$ elements, then we are done. Otherwise, we iterate over the remaining elements (in the increasing order of rank by $\pi$) and add them in $P$ as long as for each group $G_i$, $|P \cap G_i| \leq \lceil \beta_i k \rceil$

(each group has at most $\lceil \beta_i k \rceil$ elements in $P$) until the size of $P$ becomes exactly $k$. Then we use the relative ordering of the elements in $P$ as in the input ranking $\pi$ and make it the $k$-length prefix of the output ranking $\sigma$. Fill the last $d - k$ positions of $\sigma$ by the remaining elements ($[d] \setminus P$) by following their relative ordering as in the input $\pi$. See Algorithm 1 in the appendix for the pseudocode of the algorithm.

By the construction of set $P$, at the end, for each group $G_i$, $\lfloor \alpha_i k \rfloor \leq |P \cap G_i| \leq \lceil \beta_i k \rceil$. Since we use the elements of $P$ in the $k$-length prefix of the output ranking $\sigma$, $\sigma$ is an $(\bar{\alpha}, \bar{\beta})$-weak $k$-fair ranking. For the running time, a straightforward implementation our algorithms takes $\mathcal{O}(d)$ time. It only remains to argue that $\sigma$ is a closest $(\bar{\alpha}, \bar{\beta})$-weak $k$-fair ranking to the input $\pi$. To show that, we use the following key observation.

**Claim 3.3.** *Under the Kendall tau distance, there always exists a closest $(\bar{\alpha}, \bar{\beta})$- $k$-fair ranking $\pi^*$ such that, for each group $G_i$ ($i \in [g]$), for any two elements $a \neq b \in G_i$, $a <_{\pi^*} b$ if and only if $a <_\pi b$.*

We say that a ranking $\pi^*$ satisfying the above property (i.e., for each group $G_i$, $i \in [g]$, for any two elements $a \neq b \in G_i$, $a <_{\pi^*} b$ if and only if $a <_\pi b$), *preserves intra-group orderings*. This is because for elements of any group, their ordering in $\pi^*$ is the same as their ordering in $\pi$.

We want to highlight that the above claim holds for both the notions of fairness we consider. We defer the proof of the above claim and how we use it to conclude the proof of Theorem 3.2, to the appendix.

**Extension to general fairness notion.** Previously, we provide an algorithm that outputs a weak fair ranking (see Definition 2.5 for the definition of weak fairness) closest to the input. Now, we present an algorithm that outputs a closest fair ranking (according to Definition 2.4).

**Theorem 3.4.** *There exists an $\mathcal{O}(d^2)$ time algorithm that, given a ranking $\pi \in \mathcal{S}_d$, a partition of $[d]$ into $g$ groups $G_1, \ldots, G_g$ for some $g \in \mathbb{N}$, and $\bar{\alpha} = (\alpha_1, \ldots, \alpha_g) \in [0, 1]^g$, $\bar{\beta} = (\beta_1, \ldots, \beta_g) \in [0, 1]^g$, $k \in [d]$, outputs a closest $(\bar{\alpha}, \bar{\beta})$-$k$-fair ranking under the Kendall tau distance.*

The main challenge with this stronger fairness notion is that now we need to satisfy the fairness criteria for all the prefixes not just a fixed $k$-length prefix as in case of weak fairness. Surprisingly, we show that under the Kendall tau metric, by iteratively applying the algorithm for the closest weak fair ranking (Algorithm 1) as a black-box, over the prefixes of decreasing length, we can construct a closest fair ranking (not just a closest weak fair ranking). Here, it is worth noting that at any iteration the input to Algorithm 1 is a prefix of $\pi$ which may not be a permutation. However, Algorithm 1 only treats the input as a sequence of numbers (not really as a permutation). See Algorithm 2 in the appendix for a formal description of the algorithm.

Note that, since we iteratively apply Algorithm 1 on a prefix of $\pi$ (not the whole sequence represented by $\pi$), it is not even clear whether the algorithm finally outputs a fair ranking (assuming it exists). Below we first argue that if there exists a fair ranking, then the output $\sigma$ must be a fair ranking. Next, we establish that $\sigma$ is indeed a closest fair ranking to $\pi$.

Let $\pi^*$ be a closest fair ranking to $\pi$ that preserves intra-group orderings (Claim 3.3 guarantees such a closest fair ranking). We show that the output $\sigma = \pi^*$. We start the argument by considering any two prefixes of length $k_1$ and $k_2$, where $k_2 < k_1$. We argue that $k_1$ and $k_2$-length prefixes of $\sigma$ and $\pi^*$ are the same. Since this holds for any $k_1$ and $k_2$ (with $k_1, k_2 \geq k$), where $k_2 < k_1$, by using induction we can show that $\sigma = \pi^*$. We defer the induction argument to the appendix and below provide the argument for the $k_1$ and $k_2$-length prefixes (which is a key to prove the correctness of Theorem 3.4).

For the sake of analysis, let us consider the following three permutations. Let $\pi_1$ be the $(\bar{\alpha}, \bar{\beta})$-weak $k_1$-fair ranking closest to $\pi$, output by Algorithm 1. Let $\pi_2$ be the ranking output by Algorithm 1 when given the $k_1$-length prefix of $\pi_1$ (i.e., the sequence $\pi_1([k_1])$) as input and is asked to output an $(\bar{\alpha}, \bar{\beta})$-weak $k_2$-fair ranking closest to $\pi_1$. Further, let $\pi_2'$ be the $(\bar{\alpha}, \bar{\beta})$-weak $k_2$-fair ranking closest to $\pi$, output by Algorithm 1. In other words, $\pi_2$ is the ranking produced by first applying Algorithm 1 on $\pi$ to make its $k_1$-length prefix fair and then apply Algorithm 1 again on that output to make its $k_2$-length prefix fair. Whereas, $\pi_2'$ is the ranking produced by directly applying Algorithm 1 on $\pi$ to make its $k_2$-length prefix fair.

From the definition it is not at all clear whether such a $\pi_2$ even exists. The next claim argues about the existence of ranking $\pi_2$.

**Claim 3.5.** *If there is a ranking $\pi'$ such that its $k_1$-length prefix $P_1$ and $k_2$-length prefix $P_2$ satisfies that for each group $G_i$ $(i \in [g])$, $\lfloor \alpha_i k_1 \rfloor \leq |P_1 \cap G_i| \leq \lceil \beta_i k_1 \rceil$ and $\lfloor \alpha_i k_2 \rfloor \leq |P_2 \cap G_i| \leq \lceil \beta_i k_2 \rceil$, then $\pi_2$ exists.*

It can further be shown that,

**Claim 3.6.** *The set of elements in $\pi_2([k_1])$ is the same as that in $\pi_1([k_1])$.*

**Claim 3.7.** *The set of elements in $\pi_2([k_2])$ is the same as that in $\pi_2'([k_2])$.*

**Claim 3.8.** *The set of elements in $\pi^*([k_1])$ is the same as that in $\pi_2([k_1])$.*

**Claim 3.9.** *The set of elements in $\pi^*([k_2])$ is the same as that in $\pi_2([k_2])$.*

The proof of the above claims is in the appendix. We apply Claim 3.8 and Claim 3.9 iteratively to complete the correctness of Algorithm 2 which we defer to the appendix.

It only remains to argue that the algorithm runs in time $\mathcal{O}(d^2)$. This is easy to see since the algorithm invokes at most $d$ calls to the $\mathcal{O}(d)$ subroutine Algorithm 1.

### 3.2 Closest fair ranking under Ulam metric

**Theorem 3.10.** *There exists a polynomial time dynamic programming based algorithm that finds a $(\bar{\alpha}, \bar{\beta})$-$k$-fair ranking under Ulam metric when there are constant number of groups.*

The proof of the lemma uses an intricate dynamic program exploiting the connection between the Ulam distance with the Longest Common Subsequence problem. We defer the proof to the appendix.

## 4 Fair Rank Aggregation

We start this section by formally defining the *fair rank aggregation* problem. Then we will provide two meta-algorithms that approximate the fair aggregated ranking.

**Definition 4.1** ($q$-mean Rank Aggregation)**.** Consider a metric space $(\mathcal{S}_d, \rho)$ for a $d \in \mathbb{N}$. Given an exponent parameter $q \in \mathbb{R}$, and a set $S \subseteq \mathcal{S}_d$ of $n$ input rankings, the *$q$-mean rank aggregation* problem asks to find a ranking $\sigma \in \mathcal{S}_d$ (not necessarily from $S$) that minimizes the objective function $\text{Obj}_q(S, \sigma) := \left( \sum_{\pi \in S} \rho(\pi, \sigma)^q \right)^{1/q}$.

Generalized mean or $q$-mean objective functions are well-studied in the context of clustering [CMV], and division of goods [BKM22]. We study it for the first time in the context of rank aggregation. For $q = 1$, the above problem is also referred to as the *median ranking* problem or simply *rank aggregation* problem [Kem59, You88, YL78, DKNS01]. On the other hand, for $q = \infty$, the problem is also referred to as the *center ranking* problem or *maximum rank aggregation* problem [BBGH15, BBD09, Pop07]. Both these special cases are studied extensively in the literature with different distance measures, e.g., Kendall tau distance [DKNS01, ACN08, KMS07, Sch12, BBD09], Ulam distance [CDK21, BBGH15, CGJ21], Spearman footrule distance [DKNS01, BBGH15].

In the fair rank aggregation problem, we want the output aggregated rank to satisfy certain fairness constraints. Throughout this section, for brevity, we use the term (weak) fair ranking instead of $(\bar{\alpha}, \bar{\beta})$-(weak) $k$-fair ranking.

**Definition 4.2** ($q$-mean Fair Rank Aggregation)**.** Consider a metric space $(\mathcal{S}_d, \rho)$ for a $d \in \mathbb{N}$. Given an exponent parameter $q \in \mathbb{R}$, and a set $S \subseteq \mathcal{S}_d$ of $n$ input rankings/permutations, the *$q$-mean (weak) fair rank aggregation* problem asks to find a (weak) fair ranking $\sigma \in \mathcal{S}_d$ (not necessarily from $S$) that minimizes the objective function $\text{Obj}_q(S, \sigma) := \left( \sum_{\pi \in S} \rho(\pi, \sigma)^q \right)^{1/q}$.

It is worth noting that in the above definition, the minimization is over the set of all the (weak) fair rankings in $\mathcal{S}_d$. When clear from the context, we drop weak and refer to it as the $q$-mean fair rank aggregation problem. Let $\sigma^*$ be a (weak) fair ranking that minimizes $\text{Obj}_q(S, \sigma)$, then we call $\sigma^*$ a *$q$-mean fair aggregated rank* of $S$. We refer to $\text{Obj}_q(S, \sigma^*)$ as $\text{OPT}_q(S)$.

When $q = 1$, we refer the problem as the *fair median ranking* problem or simply *fair rank aggregation* problem. When $q = \infty$, the objective function becomes $\text{Obj}_\infty(S, \sigma) = \max_{\pi \in S} \rho(\pi, \sigma)$, and we refer the problem as the *fair center ranking* problem.

Next, we present two meta algorithms that work for any values of $q$ and irrespective of weak or strong (i.e., general) fairness constraint. We will also assume $q$ to be a constant.

## 4.1 First Meta Algorithm

**Theorem 4.3.** *Consider any $q \geq 1$. Suppose there is a $t(d)$-time $c$-approximation algorithm $\mathcal{A}$, for some $c \geq 1$, for the closest fair ranking problem over the metric space $(\mathcal{S}_d, \rho)$. Then there exists a $(c+2)$-approximation algorithm for the $q$-mean fair rank aggregation problem, that runs in $\mathcal{O}(n \cdot t(d) + n^2 \cdot f(d))$ time where $f(d)$ is the time to compute $\rho(\pi_1, \pi_2)$ for any $\pi_1, \pi_2 \in \mathcal{S}_d$.*

We devote this subsection to proving the above theorem. Let us start by describing the algorithm. It works as follows: Given a set $S \subseteq \mathcal{S}_d$ of rankings, it first computes $c$-approximate closest fair ranking $\sigma$ (for some $c \geq 1$) for each $\pi \in S$. Next, among these $|S|$ fair rankings output the ranking $\sigma$ that minimizes $\mathtt{Obj}_q(S, \sigma)$. Let us denote the output ranking by $\bar{\sigma}$. See Algorithm 4 in the appendix for a more formal description.

It is straightforward to verify that the running time of the above algorithm is $\mathcal{O}(n \cdot t(d) + n^2 \cdot f(d))$, where $f(d)$ is the time to compute $\rho(\pi_1, \pi_2)$ for any $\pi_1, \pi_2 \in \mathcal{S}_d$ and $t(d)$ denotes the running time of the algorithm $\mathcal{A}$. So it only remains to argue about the approximation factor of Algorithm 4. The following simple observation plays a pivotal role in establishing the approximation factor of Algorithm 4.

**Lemma 4.4.** *Given a set $S \subseteq \mathcal{S}_d$ of $n$ rankings, let $\sigma^*$ be an optimal $q$-mean fair aggregated rank of $S$ under a distance function $\rho$. Further, let $\bar{\pi}$ be a nearest neighbor (closest ranking) of $\sigma^*$ in $S$, and $\bar{\sigma}$ be a $c$-approximate closest fair ranking to $\bar{\pi}$, for some $c \geq 1$. Then $\forall \pi \in S, \rho(\pi, \bar{\sigma}) \leq (c+2) \cdot \rho(\pi, \sigma^*)$.*

We defer the proof of the above claim to the appendix. Now, we use the above lemma to show that the approximation factor of Algorithm 4 is $c + 2$. Let $\sigma^*$ be an (arbitrary) optimal fair aggregate rank of $S$ and $\bar{\sigma}$ be the output of Algorithm 4. The optimal value of the objective function is $\mathtt{OPT} = \mathtt{Obj}_q(S, \sigma^*) = \left( \sum_{\pi \in S} \rho(\pi, \sigma^*)^q \right)^{1/q}$. Next, we show that $\mathtt{Obj}_q(S, \bar{\sigma}) \leq (c+2) \cdot \mathtt{OPT}$.

$$\mathtt{Obj}_q(S, \bar{\sigma}) \leq \left( \sum_{\pi \in S} \left( (c+2) \cdot \rho(\pi, \sigma^*) \right)^q \right)^{1/q} = (c+2) \cdot \left( \sum_{\pi \in S} \rho(\pi, \sigma^*)^q \right)^{1/q} = (c+2) \cdot \mathtt{OPT}.$$

where the first inequality follows from Lemma 4.4. This concludes the proof of Theorem 4.3.

**Applications of Theorem 4.3.** We have shown in Theorem 3.4 that the closest fair ranking problem for Kendall tau can be solved exactly in $\mathcal{O}(d^2)$ time, i.e., the approximation ratio is $c = 1$. We also know from [Kni66], that the Kendall tau distance between two permutations can be computed in $\mathcal{O}(d \log d)$ time. This gives us that,

**Corollary 4.5.** *For any $q \geq 1$, there exists an $\mathcal{O}(nd^2 + n^2 d \log d)$ time meta-algorithm that finds a 3-approximate solution to the $q$-mean fair rank aggregation problem under the Kendall tau metric.*

It is shown in [CSV18] that the closest fair ranking problem for Spearman footrule can be solved exactly in $\mathcal{O}(d^3 \log d)$ time, i.e., the approximation ratio is $c = 1$. Since distance under Spearman footrule can be trivially computed in $\mathcal{O}(d)$ we have that,

**Corollary 4.6.** *For any $q \geq 1$, there exists an $\mathcal{O}(nd^3 \log d + n^2 d)$ time meta-algorithm that finds a 3-approximate solution to the $q$-mean fair rank aggregation problem under the Spearman footrule metric.*

We have shown in Theorem 3.10 that for constant number of groups, the closest fair ranking problem for Ulam metric can be solved exactly in $\mathcal{O}(d^{g+2})$ time, i.e., the approximation ratio is $c = 1$. From [AD99] we know that Ulam distance between two permutations can be computed in $\mathcal{O}(d \log d)$ time. This gives us that,

**Corollary 4.7.** *For any $q \geq 1$, there exists an $\mathcal{O}(nd^{g+2} + n^2 d \log d)$ time meta-algorithm, that finds a 3-approximate solution to the $q$-mean fair rank aggregation problem , under the Ulam metric.*

We would like to emphasize that all the above results hold for any values of $q \geq 1$. Hence, they are also true for the special case of the fair median problem (i.e., for $q = 1$) and the fair center problem (i.e., for $q = \infty$).

## 4.2 Second Meta Algorithm

**Theorem 4.8.** *Consider any $q \geq 1$. Suppose there is a $t_1(n)$ time $c_1$-approximation algorithm $\mathcal{A}_1$ for some $c_1 \geq 1$ for q-mean rank aggregation problem; and a $t_2(d)$-time $c_2$-approximation algorithm $\mathcal{A}_2$, for some $c_2 \geq 1$, for the closest fair ranking problem over the metric space $(\mathcal{S}_d, \rho)$. Then there exists a $(c_1 c_2 + c_1 + c_2)$-approximation algorithm for the q-mean fair rank aggregation problem, that runs in $\mathcal{O}(t_1(n) + t_2(d) + n^2 \cdot f(d))$ time where $f(d)$ is the time to compute $\rho(\pi_1, \pi_2)$ for any $\pi_1, \pi_2 \in \mathcal{S}_d$.*

The algorithm works as follows: Given a set $S \subseteq \mathcal{S}_d$ of rankings, it first computes $c_1$-approximate aggregate rank $\pi^*$. Next, output a $c_2$-approximate closest fair ranking $\bar{\sigma}$, to $\pi^*$. See Algorithm 5 in the appendix for a more formal description.

It is easy to see that the running time of the algorithm is $\mathcal{O}(t_1(n) + t_2(d) + n^2 \cdot f(d))$, where $f(d)$ is the time to compute $\rho(\pi, \sigma)$ for any $\pi, \sigma \in \mathcal{S}_d$, $t_1(n)$ denotes the running time of the algorithm $\mathcal{A}_1$, and $t_2(d)$ denotes the running time of the algorithm $\mathcal{A}_2$. It now remains to argue about the approximation ratio of the above algorithm. We again make a simple but crucial observation towards establishing the approximation ratio for Algorithm 5.

**Lemma 4.9.** *Given a set $S \subseteq \mathcal{S}_d$ of $n$ rankings, let $\sigma^*$ be an optimal q-mean fair aggregated rank of $S$ under a distance function $\rho$. Further, let $\pi^*$ be the $c_1$-approximate aggregate rank of $S$ and $\bar{\sigma}$ be a $c_2$-approximate closest fair ranking to $\pi^*$, for some $c_1, c_1 \geq 1$. Then*

$$\forall \pi \in S, \rho(\pi, \bar{\sigma}) \leq (c_1 c_1 + c_1 + c_2) \cdot \rho(\pi, \sigma^*).$$

We defer the proof of this lemma to the appendix. Once we have this key lemma in place, the remaining proof of Theorem 4.8, follows exactly as the proof of Theorem 4.3.

The above algorithm can give similar approximation guarantees as Algorithm 4, but with potentially better running times depending on whether the rank aggregation problem is solved in a faster way for the particular problem in consideration. For instance consider the case for Spearman footrule. It is known that the rank aggregation problem for Spearman footrule can be solved in $\tilde{\mathcal{O}}(d^2)$ time [vdBLN+20]. So, using this in conjunction with Algorithm 5 we obtain the following result.

**Corollary 4.10.** *For $q = 1$, there exists an $\mathcal{O}(d^3 \log d + n^2 d + nd^2)$ time meta-algorithm, that finds a 3-approximate solution to the q-mean fair rank aggregation problem (i.e., the fair median problem) under Spearman footrule metric.*

## 4.3 Breaking below 3-factor for Ulam

**Theorem 4.11.** *For $q = 1$, there exists a constant $\varepsilon > 0$ and a polynomial time algorithm that finds a $(3 - \varepsilon)$-approximate solution to the q-mean fair rank aggregation problem (i.e., the fair median problem), under the Ulam metric for constantly many groups.*

We show the above result by designing an algorithm based on the relative ordering of the elements (as in the majority of the input rankings). Then the final output is the best of that output by this new algorithm and that produced by our first meta-algorithm. We argue that when the whole optimal objective value is distributed among only a few elements, then the first meta-algorithm already achieves $(3 - \varepsilon)$-approximation. Otherwise, this relative ordering-based approach will provide a $(3 - \varepsilon)$-approximation. The argument used is quite similar to that used in [CDK21] to obtain a better than 2-factor approximate median under the Ulam metric. We describe our algorithm along with the complete analysis in the appendix.

## 5 Conclusion

In this paper, we lay the theoretical framework for studying the closest fair ranking and fair rank aggregation problems while ensuring minority protection and restricted dominance. We first give a simple, practical, and exact algorithm for CFR under the Kendall tau metric; and a polynomial time exact algorithm for CFR under the Ulam metric when there are constantly many groups. We then use such a black box solution to CFR to design two novel meta-algorithms for FRA for a general $q$-mean objective, which are valid under any metric. The approximation ratios of these meta algorithms depend on the approximation ratio of the CFR subroutine used by them. Lastly, we

give a $(3 - \varepsilon)$-approximate algorithm for FRA under the Ulam metric, which improves over our meta algorithm's approximation ratio for the same case. Achieving a similar (better than 3-factor) approximation bound for the Kendall tau or other metrics is an interesting open problem. Another set of intriguing open problems arise when there is overlap between the groups, i.e., when an element can belong to multiple groups.

It is still open whether exact algorithms exist for the the case when an element can belong to (at most) two groups.

**Acknowledgments.** Diptarka Chakraborty was supported in part by an MoE AcRF Tier 2 grant (WBS No. A-8000416-00-00) and an NUS ODPRT grant (WBS No. A-0008078-00-00). Arindam Khan gratefully acknowledges the generous support due to Pratiksha Trust Young Investigator Award, Google India Research Award, and Google ExploreCS Award.

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
