# A Omitted Proofs

## A.1 Closest Fair Ranking: Kendall tau

**For weak fairness.** We start with describing the pseudocode of our algorithm.

---

**Algorithm 1:** Algorithm to compute closest weak fair ranking under Kendall tau

---

**Input:** Input ranking $\pi \in \mathcal{S}_d$, $g$ groups $G_1, \ldots, G_g$, $\bar{\alpha} = (\alpha_1, \ldots, \alpha_g) \in [0,1]^g$,
$\quad$ $\bar{\beta} = (\beta_1, \ldots, \beta_g) \in [0,1]^g$, $k \in [d]$.
**Output:** An $(\bar{\alpha}, \bar{\beta})$-weak $k$-fair ranking.

1 Initialize a set $P \leftarrow \emptyset$.
2 Initialize a ranking $\sigma \leftarrow \emptyset$.
3 From each group $i \in [g]$, pick the top $\lfloor \alpha_i k \rfloor$ elements of $G_i$ according to the ranking $\pi$ and add them to $P$.
4 **if** $|P| > k$ **then**
5 $\quad$ **return** *No fair ranking exists.*

6 **else**
7 $\quad$ Iterate over the remaining elements (in increasing order of rank by $\pi$) and add them in $P$ as long as $|P| \le k$ and for each $i \in [g]$, $|P \cap G_i| \le \lceil \beta_i k \rceil$.
8 $\quad$ $\sigma \leftarrow$ Construct a new ranking by first placing the elements of $P$ in the top $k$ positions (i,e., in the $k$-length prefix) while preserving the relative ordering among themselves according to $\pi$, and then placing the remaining elements ($[d] \setminus P$) in the rest (the last $d - k$ positions) while preserving the relative ordering among themselves according to $\pi$.

9 Iterate over ranking $\sigma$, and count the fraction of elements in the top-$k$ from each group.
10 **if** $\sigma$ is $(\bar{\alpha}, \bar{\beta})$-*weak $k$-fair* **then**
11 $\quad$ **return** $\sigma$.

12 **else**
13 $\quad$ **return** *No fair ranking exists.*

---

In the algorithm description above, for a subset $P \subseteq [d]$, we say that an ordering $\sigma$ of the elements of $P$ *preserves relative orderings according to $\pi$*, when for any two elements $a \ne b \in P$, $a <_\sigma b$ if and only if $a <_\pi b$.

We now show that under Kendall tau, there exists an optimal $(\bar{\alpha}, \bar{\beta})$ - $k$-fair ranking that preserves all intra-group orderings from the input permutation.

**Claim 3.3.** *Under the Kendall tau distance, there always exists a closest $(\bar{\alpha}, \bar{\beta})$- $k$-fair ranking $\pi^*$ such that, for each group $G_i$ ($i \in [g]$), for any two elements $a \ne b \in G_i$, $a <_{\pi^*} b$ if and only if $a <_\pi b$.*

*Proof.* Let $\pi$ be the input ranking, and $\pi'$ be some closest $(\bar{\alpha}, \bar{\beta})$-$k$-fair ranking. If there exists $x, y \in G$ for some group $G$, such that $x <_\pi y$ but $x >_{\pi'} y$, then we claim that the permutation $\pi''$ obtained by swapping $x$ and $y$, is fair, and satisfies $\mathcal{K}(\pi, \pi'') \le \mathcal{K}(\pi, \pi')$. Hence, by a series of such swap operations, any $(\bar{\alpha}, \bar{\beta})$-$k$-fair ranking $\pi'$, can be transformed into one that satisfies the property that, $a \ne b \in G_i$, $a <_{\pi'} b$ if and only if $a <_\pi b$.

We partition the set $[d] \setminus \{x, y\}$ into three sets, $L = \{z : z <_{\pi''} x\}$, $B = \{z : x <_{\pi''} z <_{\pi''} y\}$, $U = \{z : y <_{\pi''} z\}$. Now, consider the sets $K_{\pi'} = \{(i, j) \in [d] \times [d] \mid (i <_\pi j) \wedge (i >_{\pi'} j)\}$, $K_{\pi''} = \{(i, j) \in [d] \times [d] \mid (i <_\pi j) \wedge (i >_{\pi''} j)\}$.

Consider any element $(a, b) \in K_{\pi''}$, such that $a, b$ belong to the same group.

- If $a, b \notin \{x, y\}$: then clearly the pair of elements is in the same positions in $\pi'$ also, and hence $(a, b) \in K_{\pi'}$.

- If $y \in \{a, b\}$: let $a' \in \{a, b\} \setminus \{y\}$.

- $a' \in U$: This means that $x <_{\pi''} y <_{\pi''} a'$, so with swapped $x$ and $y$, we will have that $y <_{\pi'} x <_{\pi'} a'$. The relative position of $y$ and $a'$ remains the same and hence $(a', y) \in K_{\pi'}$.
- $a' \in B$: This means that $x <_{\pi''} a' <_{\pi''} y$, so with swapped $x$ and $y$, we will have that $y <_{\pi'} a' <_{\pi'} x$. But we know that $(y, a') \in K_{\pi''}$, which means that $x <_\pi y <_\pi a'$. So we have that $x <_\pi a' \wedge a' <_{\pi'} x$, which gives us that $(x, a') \in K_{\pi'}$.
- $a' \in L$: This means that $a' <_{\pi''} x <_{\pi''} y$, so with swapped $x$ and $y$, we will have that $a' <_{\pi'} y <_{\pi'} x$. The relative position of $y$ and $a'$ remains the same and hence $(y, a') \in K_{\pi'}$.

- If $x \in \{a, b\}$: This case can also be analyzed similarly to the case above.

Now, note that $(x, y)$ is not in $K_{\pi''}$, so we get that $|K_{\pi''}| \le |K_{\pi'}|$. Also, since this swap operation does not change the number (or fraction) of elements from any group in the top-$k$ ranks, $\pi''$ is still fair. This completes our proof. $\square$

**Theorem 3.2.** *There exists a linear time algorithm that, given a ranking $\pi \in \mathcal{S}_d$, a partition of $[d]$ into $g$ groups $G_1, \dots, G_g$ for some $g \in \mathbb{N}$, and $\bar{\alpha} = (\alpha_1, \dots, \alpha_g) \in [0,1]^g$, $\bar{\beta} = (\beta_1, \dots, \beta_g) \in [0,1]^g$, $k \in [d]$, outputs a closest $(\bar{\alpha}, \bar{\beta})$-weak $k$-fair ranking under the Kendall tau distance.*

*Proof.* Let $\pi$ be the given input ranking, $\pi_{\mathsf{OPT}}$ be a closest ranking that also preserves intra-group orderings (the existence of which is guaranteed by Claim 3.3), and $\pi_{\mathsf{GRD}}$ be the ranking returned by Algorithm 1. First, observe, by the construction of $\pi_{\mathsf{GRD}}$ (in Algorithm 1), $\pi_{\mathsf{GRD}}$ also preserves intra-group orderings, and moreover, the relative ordering among the set of symbols in $\pi_{\mathsf{GRD}}[k]$ (and in $[d] \setminus \pi_{\mathsf{GRD}}[k]$) is the same as that in $\pi$. It is also easy to see that the relative ordering among the set of symbols in $\pi_{\mathsf{OPT}}[k]$ (and in $[d] \setminus \pi_{\mathsf{OPT}}[k]$) is the same as that in $\pi$; otherwise, by only changing the relative ordering of the set of symbols in $\pi_{\mathsf{OPT}}[k]$ (and in $[d] \setminus \pi_{\mathsf{OPT}}[k]$) to that in $\pi$, we get another fair ranking $\pi'$ such that $\mathcal{K}(\pi', \pi) < \mathcal{K}(\pi_{\mathsf{OPT}}, \pi)$, which leads to a contradiction.

Now, assume towards contradiction that the rankings $\pi_{\mathsf{OPT}}$ and $\pi_{\mathsf{GRD}}$ are not the same, which implies (from the discussion in the last paragraph) that there is a symbol $a \in \pi_{\mathsf{GRD}}[k] \setminus \pi_{\mathsf{OPT}}[k]$ and $b \in \pi_{\mathsf{OPT}}[k] \setminus \pi_{\mathsf{GRD}}[k]$. Since both $\pi_{\mathsf{OPT}}$ and $\pi_{\mathsf{GRD}}$ preserve intra-group orderings, $a$ and $b$ cannot belong to the same group, i.e., $a \in G_i$ and $b \in G_j$ such that $i \ne j$. Then since $\pi_{\mathsf{OPT}}$ is a fair ranking that preserves intra-group orderings,

1. element $a$ is not among the top $\lfloor \alpha_i k \rfloor$ elements of $G_i$ according to $\pi$; and

2. element $b$ is among the top $\lceil \beta_j k \rceil$ elements of $G_j$ according to $\pi$.

Next, we argue that $a <_\pi b$. If not, since $a <_{\pi_{\mathsf{GRD}}} b$, it follows from the construction of $\pi_{\mathsf{GRD}}$ (in Algorithm 1), either $a$ is among the top $\lfloor \alpha_i k \rfloor$ elements of $G_i$ according to $\pi$ contradicting Item 1, or $b$ is not among the top $\lceil \beta_j k \rceil$ elements of $G_j$ according to $\pi$ contradicting Item 2. Hence, we conclude that $a <_\pi b$.

We now claim that by swapping $a$ and $b$ in $\pi_{\mathsf{OPT}}$, we get a ranking $\pi'$ that is fair and also reduces the Kendall tau distance from $\pi$, giving us a contradiction. Let us start by arguing that $\pi'$ is fair. Note, $\pi'[k] = (\pi_{\mathsf{OPT}}[k] \setminus \{b\}) \cup \{a\}$. Next, observe, since $\pi_{\mathsf{GRD}}$ is a fair ranking that preserves intra-group orderings,

- element $b$ is not among the top $\lfloor \alpha_j k \rfloor$ elements of $G_j$ according to $\pi$; and

- element $a$ is among the top $\lceil \beta_i k \rceil$ elements of $G_i$ according to $\pi$.

As a consequence, swapping $a$ and $b$ in $\pi_{\mathsf{OPT}}$ to get $\pi'$ does not violate any of the fairness constraints for the groups $G_i$ and $G_j$ (and, of course, none of the other groups). Hence, $\pi'$ is also a fair ranking. So it remains to argue that $\mathcal{K}(\pi', \pi) < \mathcal{K}(\pi_{\mathsf{OPT}}, \pi)$ to derive the contradiction.

Observe that the set of pair of symbols for which the relative ordering changes in $\pi'$ as compared to $\pi_{\mathsf{OPT}}$ is the following:
$$\{(b, c), (c, a) \mid b <_{\pi_{\mathsf{OPT}}} c <_{\pi_{\mathsf{OPT}}} a\}.$$

Consider any element $c$ such that $b <_{\pi_{\mathsf{OPT}}} c <_{\pi_{\mathsf{OPT}}} a$. Note, $a <_{\pi'} c <_{\pi'} b$.

- If the pair $(c, a)$ creates an inversion (with respect to $\pi$) in $\pi'$, but not in $\pi_{\mathtt{OPT}}$, then $c <_\pi a$. Now, since $a <_\pi b$, we have $c <_\pi b$. Thus the pair $(b, c)$ creates an inversion (with respect to $\pi$) in $\pi_{\mathtt{OPT}}$, but not in $\pi'$.

- If the pair $(b, c)$ creates an inversion (with respect to $\pi$) in $\pi'$, but not in $\pi_{\mathtt{OPT}}$, then $b <_\pi c$. Now, since $a <_\pi b$, we have $a <_\pi c$. Thus the pair $(c, a)$ creates an inversion (with respect to $\pi$) in $\pi_{\mathtt{OPT}}$, but not in $\pi'$.

Hence, we see that $\pi'$ has at least one lesser inverted pair (being $a$ and $b$) than $\pi_{\mathtt{OPT}}$. In other words, $\mathcal{K}(\pi', \pi) < \mathcal{K}(\pi_{\mathtt{OPT}}, \pi)$, leading to a contradiction, which completes the proof. $\square$

**For general fairness notion.**  We first provide the pseudocode of our algorithm.

---
**Algorithm 2:** Algorithm to compute closest fair ranking under Kendall tau

---
**Input:** Input ranking $\pi \in \mathcal{S}_d$, $g$ groups $G_1, \ldots, G_g$, $\bar{\alpha} = (\alpha_1, \ldots, \alpha_g) \in [0, 1]^g$,
$\quad$ $\bar{\beta} = (\beta_1, \ldots, \beta_g) \in [0, 1]^g$, $k \in [d]$.
**Output:** An $(\bar{\alpha}, \bar{\beta})$-$k$-fair ranking.
1 Start with the largest prefix (i.e., the whole $d$-length prefix) $P$ of $\pi$, and check whether for each
$\quad$ $G_i$, $\lfloor \alpha_i |P| \rfloor \leq |P \cap G_i| \leq \lceil \beta_i |P| \rceil$. If not, return "No fair ranking exists"; else continue.
2 **for** $p = d - 1$ *to* $p = k - 1$ **do**
3 $\quad$ Let $P = \pi([p+1])$ be the $p + 1$-length prefix of $\pi$.
4 $\quad$ $\pi \leftarrow$ Algorithm 1 $(\pi, (G_1, \ldots, G_g), \bar{\alpha}, \bar{\beta}, p)$.
5 **return** *The resulting ranking stored in variable $\pi$.*

---

**Claim 3.5.** *If there is a ranking $\pi'$ such that its $k_1$-length prefix $P_1$ and $k_2$-length prefix $P_2$ satisfies that for each group $G_i$ ($i \in [g]$), $\lfloor \alpha_i k_1 \rfloor \leq |P_1 \cap G_i| \leq \lceil \beta_i k_1 \rceil$ and $\lfloor \alpha_i k_2 \rfloor \leq |P_2 \cap G_i| \leq \lceil \beta_i k_2 \rceil$, then $\pi_2$ exists.*

*Proof.* Our proof is constructive. Since $\pi'$ exists, $\pi_1$ also exists (follows from the correctness of Algorithm 1). Since for any group $G_i$, $|\pi_1([k_1]) \cap G_i| \geq \lfloor \alpha_i k_1 \rfloor \geq \lfloor \alpha_i k_2 \rfloor$, the elements in the prefix $\pi_1([k_1])$ are sufficient to satisfy the lower bound fairness constraints for the $k_2$-length prefix.

For a group $i \in [g]$, let $\ell_i$ be the number of items of group $G_i$ in the prefix $\pi_1([k_1])$. Next, we argue that assuming the existence of $\pi'$ (as in the claim statement), it is always possible to construct a $\pi_2$ for which the upper bound fairness constraints are satisfied for the $k_2$-length prefix. Note, $\sum_{i \in [g]} \ell_i = k_1$ and for each $i \in [g]$, $\ell_i \leq \lceil \beta_i k_1 \rceil$.

Consider $L_i'$ to be the set of top $\left\lceil \ell_i \times \frac{k_2}{k_1} \right\rceil$ elements from $\pi_1([k_1]) \cap G_i$. Let $|L_i'| = \ell_i'$. Then,

$$\sum_{i \in [g]} \ell_i' = \sum_{i \in [g]} \left\lceil \ell_i \times \frac{k_2}{k_1} \right\rceil \geq \left\lceil \frac{k_2}{k_1} \times \sum_{i \in [g]} \ell_i \right\rceil \geq k_2.$$

Also note that for each $i \in [g]$,

$$\ell_i' = \left\lceil \ell_i \times \frac{k_2}{k_1} \right\rceil \leq \left\lceil \beta_i k_1 \times \frac{k_2}{k_1} \right\rceil \leq \lceil \beta_i k_2 \rceil$$

So, by ranking the above set $L_i'$ of chosen elements, we can ensure that we can satisfy both the upper and lower bound fairness constraints for the $k_2$-prefix. Thus we get an $(\bar{\alpha}, \bar{\beta})$-weak $k_2$-fair ranking whose elements in the $k_2$-prefix are from the set $\pi_1([k_1])$. Since one such valid solution exists, Algorithm 1 can find an appropriate valid solution $\pi_2$. $\square$

**Claim 3.6.** *The set of elements in $\pi_2([k_1])$ is the same as that in $\pi_1([k_1])$.*

*Proof.* We only call the Algorithm 1 subroutine on the $k_1$-length prefix of $\pi_1$ to construct $\pi_2$. So by construction, we have that for the $k_1$-length prefix, the set of elements in both the permutations are the same. $\square$

**Claim 3.7.** *The set of elements in $\pi_2([k_2])$ is the same as that in $\pi_2'([k_2])$.*

*Proof.* Consider an element $a \in \pi_2'([k_2]) \cap G_i$ for some $i \in [g]$. If $a$ is among the top $\lfloor \alpha_i k_2 \rfloor$ elements (according to $\pi$) inside the group $G_i$, then by Algorithm 1, it would also be selected in $\pi_1([k_1])$ (since $k_1 \geq k_2$) and also in $\pi_2([k_2])$.

Now consider the case where $a$ is among the top $\lceil \beta_i k_2 \rceil$ elements of $G_i$, but not among the top $\lfloor \alpha_i k_2 \rfloor$ elements. This means that $a$ is also among the top $\lceil \beta_i k_1 \rceil (\geq \lceil \beta_i k_2 \rceil)$ elements of its group $G_i$. This means that if it is encountered during the execution of Algorithm 1 on $\pi$ to get an $(\bar{\alpha}, \bar{\beta})$-weak $k_1$-fair ranking, then it will be selected in $\pi_1([k_1])$. However, we also know that it is selected in $\pi_2'([k_2])$ which is a shorter prefix. Since the upper bound constraints were not violated for $G_i$ during the selection of the elements in $\pi_2'([k_2])$, the upper bound constraints cannot be violated during the selection of the elements of $\pi_1([k_1])$ as well. Hence, $a$ will be selected in $\pi_1([k_1])$.

By a similar argument, when executing Algorithm 1 on $\pi_1$ (in the later iteration) to output an $(\bar{\alpha}, \bar{\beta})$-weak $k_2$-fair ranking, $a$ will again be encountered and be selected in $\pi_2([k_2])$. Therefore, every element in $\pi_2'([k_2])$ is also in $\pi_2([k_2])$. Since the sizes of both the sets are equal, the two sets are in fact the same, and so are the rankings (by Algorithm 1). $\qquad\square$

**Claim 3.8.** *The set of elements in $\pi^*([k_1])$ is the same as that in $\pi_2([k_1])$.*

*Proof.* Assume towards contradiction that there exists $a \in \pi_2([k_1]) \setminus \pi^*([k_1])$ and $b \in \pi^*([k_1]) \setminus \pi_2([k_1])$. If $a, b$ were in the same group, then by Algorithm 1, we know that $a <_\pi b$, and hence by swapping the elements in $\pi^*$, the distance from $\pi$ can only be reduced (as shown in latter part of proof of Theorem 3.2). Hence we can obtain a different solution $\bar{\pi}$ in which $a \in \bar{\pi}([k_1])$ and $b \notin \bar{\pi}([k_1])$, and this is also fair. This contradicts that $\pi^*$ is a closest fair ranking to $\pi$ that preserves intra-group orderings.

In the other case, $a$ and $b$ are not in the same group, i.e., $a \in G_i$ and $b \in G_j$ for some $i \neq j$. Now we note that $a$ cannot be among the top $\lfloor \alpha_i k_1 \rfloor$ elements, but is in the top $\lceil \beta_i k_1 \rceil$ elements in $G_i$. Similarly, $b$ cannot be among the top $\lfloor \alpha_j k_1 \rfloor$ elements, but is in the top $\lceil \beta_j k_1 \rceil$ elements in $G_j$. Again, it follows from Theorem 3.2, $a <_\pi b$, and that by swapping these two elements in $\pi^*$ we can only reduce the distance from $\pi$, while obtaining another fair ranking. This again contradicts that $\pi^*$ is a closest fair ranking to $\pi$ that preserves intra-group orderings. The claim now follows. $\qquad\square$

**Claim 3.9.** *The set of elements in $\pi^*([k_2])$ is the same as that in $\pi_2([k_2])$.*

*Proof.* From Claim 3.7 we know that $\pi_2([k_2]) = \pi_2'([k_2])$. So, it suffices to prove that the sets $\pi^*([k_2])$ and $\pi_2'([k_2])$ are equal. Note that this amounts to showing that for some prefix, the output of Algorithm 1 and the optimal solution have the same set of elements. The proof is hence very similar to that of Claim 3.8. $\qquad\square$

**Theorem 3.4.** *There exists an $\mathcal{O}(d^2)$ time algorithm that, given a ranking $\pi \in \mathcal{S}_d$, a partition of $[d]$ into $g$ groups $G_1, \ldots, G_g$ for some $g \in \mathbb{N}$, and $\bar{\alpha} = (\alpha_1, \ldots, \alpha_g) \in [0, 1]^g$, $\bar{\beta} = (\beta_1, \ldots, \beta_g) \in [0, 1]^g$, $k \in [d]$, outputs a closest $(\bar{\alpha}, \bar{\beta})$-$k$-fair ranking under the Kendall tau distance.*

*Proof.* We show by induction that Algorithm 2 in fact outputs the same ranking (referred to as the greedy solution) as the optimal fair ranking $\pi^*$, which preserves intra-group orderings. In the induction we consider a prefix, at the step at which subroutine Algorithm 1 was executed on it.

**Hypothesis:** After Algorithm 2 calls the subroutine on a prefix of length $n - i$, for some $i < n - k$, the result is the optimal $(\bar{\alpha}, \bar{\beta})$- $(n - i)$-fair ranking, which preserves intra-group orderings.

**Base Case:** For $i = 0$, the prefix of length $n - i = n$ is the entire array. This is already fair (we assume that a solution to the problem exists, which means that the input set of elements must satisfy fairness criteria), and also preserves intra-group relative orderings (by construction from subroutine Algorithm 1).

**Induction Step:** Let $P_2$ be the prefix of length $n - (i + 1)$ at which the subroutine Algorithm 1 was just executed. So the subroutine was executed on on a prefix $P_1$ of length $n - i$ in the previous step, and hence it is already fair (by induction hypothesis).

From Claim 3.9 and Claim 3.8 we have that both the greedy (Algorithm 2's output) and the optimal solution have the same set of elements in both the top $P_1$ and $P_2$ prefixes. And we know that the optimal solution (by definition) and the greedy solution (by construction) preserve relative orderings w.r.t. $\pi$. This implies that the greedy solution $\pi_2$ is in fact the same as the optimal solution $\pi^*$. $\qquad\square$

## A.2 Closest Fair Ranking: Ulam

---

**Algorithm 3:** DP algorithm for $(\bar{\alpha}, \bar{\beta})$-k-fair ranking under Ulam

---

**Input:** Input ranking $\pi$, $g$ groups $G_1, \ldots, G_g$, vectors
$\quad\quad \bar{\alpha} = (\alpha_1, \ldots, \alpha_g) \in [0,1]^g, \bar{\beta} = (\beta_1, \ldots, \beta_g) \in [0,1]^g, k \in [d]$.
**Output:** An $(\bar{\alpha}, \bar{\beta})$-k-fair ranking.

1  $DP[d][|G_1|] \ldots [|G_g|] := \vec{0}$.
2  $\sigma[d][|G_1|] \ldots [|G_g|] := \emptyset$.
3  **for** $j \in [d]$ **do**                                             /* Base cases */
4     **for** $i \in [g]$ **do**
5        $DP[j][0] \ldots [a_i = 1] \ldots [0] = 1$ if there is an element of $G_i$ in $\pi[1 \ldots j]$.
6        $\sigma[j][0] \ldots [a_i = 1] \ldots [0] =$ first element of group $G_i$.

7  **Function** FnDP$([x][y_1] \ldots [y_g])$**:**
8     **if** $DP[x][y_1 \ldots [y_g] \neq 0$ **then**
9        **return** $DP[x][y_1] \ldots [y_g]$.

10    $\ell := \sum_{z \in [g]} y_z$.
11    **for** $z \in [g]$ **do**                                     /* DP recurrence loop */
12       **if** $(\ell \geq k) \, AND \, (\lfloor \alpha_z \ell \rfloor > y_z \, OR \, y_z > \lceil \beta_z \ell \rceil \, OR \, y_z > |G_z|)$ **then**
13          **return** $-\infty$.                          /* Fairness constraints */
14       **if** $\pi[x]$ *is in* $G_z$ **then**
15          **if** $DP[x-1][y_1] \ldots [y_z - 1] \ldots [y_g] + 1 > DP[x][y_1] \ldots [y_g]$ **then**
16             $DP[x][y_1] \ldots [y_g] = DP[x-1][y_1] \ldots [y_z - 1] \ldots [y_g] + 1$.
17             To obtain $\sigma[x][y_1] \ldots [y_g]$, append $\pi[x]$ to $\sigma[x-1][y_1] \ldots [y_z - 1]$.

18       **else**
19          **if** $DP[x-1][y_1] \ldots [y_z - 1] \ldots [y_g] > DP[x][y_1] \ldots [y_g]$ **then**
20             $DP[x][y_1] \ldots [y_g] = DP[x-1][y_1] \ldots [y_z - 1] \ldots [y_g]$.
21             To obtain $\sigma[x][y_1] \ldots [y_g]$, append an arbitrary element of $G_z$ to
            $\sigma[x-1][y_1] \ldots [y_z - 1]$.

22       **if** *there are more than* $y_z - 1$ *elements of* $G_z$ *in* $\pi[1 \ldots j]$ **then**
23          **if** $DP[x][y_1] \ldots [y_z - 1] \ldots [y_g] + 1 > DP[x][y_1] \ldots [y_g]$ **then**
24             $DP[x][y_1] \ldots [y_g] = DP[x][y_1] \ldots [y_z - 1] \ldots [y_g] + 1$.
25             Choose an element $g \in \{\pi[0 \ldots x]\} \setminus \{\sigma[x][y_1] \ldots [y_z - 1] \ldots [y_g]\}$.
26             Identify the first element $a$ (in increasing order of $\pi$) in the LCS, satisfying
            $g <_\pi a$ (note that $a$ could be null, representing the last position in the LCS).
27             To obtain $\sigma[x][y_1] \ldots [y_g]$: insert $g$ preceding $a$ in $\sigma[x][y_1] \ldots [y_z - 1] \ldots [y_g]$ (if
            $a$ is null, append $g$ to $\sigma[x][y_1] \ldots [y_z - 1] \ldots [y_g]$).

28       **else**
29          **if** $DP[x][y_1] \ldots [y_z - 1] \ldots [y_g] > DP[x][y_1] \ldots [y_g]$ **then**
30             $DP[x][y_1] \ldots [y_g] = DP[x][y_1] \ldots [y_z - 1] \ldots [y_g]$.
31             To obtain $\sigma[x][y_1] \ldots [y_g]$ we append an arbitrary element of $G_z$ to
            $\sigma[x][y_1] \ldots [y_z - 1]$.

32 $DP[d][|G_1|] \ldots [|G_g|] = $ FnDP$[d][|G_1|] \ldots [|G_g|]$.
33 Find the DP cell $DP[d][i_1] \ldots [i_g]$ that has the maximum value of LCS.
34 **return** $\sigma[d][i_1] \ldots [i_g]$.

---

**Theorem 3.10.** *There exists a polynomial time dynamic programming based algorithm that finds a $(\bar{\alpha}, \bar{\beta})$-$k$-fair ranking under Ulam metric when there are constant number of groups.*

*Proof.* Let $\pi$ be the given input string and $g$ be the number of groups. Let $a_i \in \mathbb{N}$, for all $i \in [g]$, and $\gamma := \sum_{i \in [g]} a_i$. Let $\mathcal{P}(a_1, \ldots, a_g)$ be the family of strings of length $\gamma$ that have exactly $a_i$ elements from group $G_i$ for all $i \in [g]$. Let $\sigma_j(a_1, \ldots, a_g)$ be the string in this family that has the longest common subsequence (LCS) with $\pi[1 \ldots j]$. So the string $\sigma_d(|G_1|, \ldots, |G_g|)$ is the string of length $d$, that has maximum LCS with $\pi$, and from the alternate definition of the Ulam metric, the smallest Ulam distance from $\pi$. So, intuitively, we define a dynamic program to compute the LCS, and build up the strings $\sigma_j(a_1, \ldots, a_g)$. The DP subproblem is defined as follows: $\mathrm{DP}[j][a_1] \ldots [a_g]$ will store the length of the longest common subsequence between $\pi[1 \ldots j]$ and $\sigma_j(a_1, \ldots, a_g)$. W.l.o.g., we also assume that it stores the string $\sigma_j(a_1, \ldots, a_g)$ as well.

With this definition, we construct a solution for the subproblem $\mathrm{DP}[j][a_1] \ldots [a_g]$ from 'smaller' subproblems as follows:

**Case 1: From subproblems corresponding to smaller values of $\gamma$.**
**Case 1A.** Subproblems due to $\pi[1 \ldots j]$: If for $i \in [g]$, the number of elements from $G_i$ in $\pi[1 \ldots j]$ is greater than $a_i - 1$ then we know that $\pi[1 \ldots j]$ has an LCS of length $\mathrm{DP}[j][a_1] \ldots [a_i - 1] \ldots [a_g]$ with $\sigma_j(a_1, \ldots, a_i - 1, \ldots, a_g)$, which does not contain all of its elements from $G_i$. So if we pick one such element $p \in G_i$, we can identify its position w.r.t. elements of the LCS in $\pi$. Let us say we identify the first element $q$ in the LCS that follows $p$ (i.e., first element in order of $\pi$ satisfying $p <_\pi a$). Then if we place $p$ right before $q$ in $\sigma_j(a_1, \ldots, a_i - 1, \ldots, a_g)$ then the LCS can now also include $p$, and hence be longer by one element.

Otherwise, if for $h \in [g]$, $\pi[1 \ldots j]$ has at most $a_h - 1$ elements from $G_h$, then $\mathrm{DP}[j][a_1] \ldots [a_h] \ldots [a_g]$ already utilizes all the elements of $G_h$ it can, and adding a new element of the group cannot extend the LCS anymore. Hence, the candidate solution's LCS size is $\mathrm{DP}[j][a_1] \ldots [a_i - 1] \ldots [a_g]$, and we can obtain this by adding an arbitrary element of $G_h$ to $\sigma_j(a_1, \ldots, a_i - 1, \ldots, a_g)$.

**Case 1B.** Subproblems due to $\pi[1 \ldots j - 1]$: For some $i \in [g]$ let $\pi[j] \in G_i$. Then one candidate solution's LCS size is $\mathrm{DP}[j-1][a_1] \ldots [a_i - 1] \ldots [a_g] + 1$, where we take the string $\sigma_{j-1}(a_1, \ldots, a_i - 1, \ldots, a_g)$ and append $\pi[j]$ at the end of it. This increases the LCS value by 1 to give a solution of size $\mathrm{DP}[j-1][a_1] \ldots [a_i - 1] \ldots [a_g] + 1$.

Otherwise, if for some $h \in [g], \pi[j] \notin G_h$, then by adding any element from $G_h$, we can never have it being the same as $\pi[j]$, and hence cannot extend the LCS any further. Hence, we can add any arbitrary element of the group to $\sigma_{j-1}(a_1, \ldots, a_h - 1, \ldots, a_g)$, to get a valid solution with LCS size $\mathrm{DP}[j-1][a_1] \ldots [a_h - 1] \ldots [a_g]$.

**Case 2: From subproblems corresponding to smaller values of $j$.** Here we try to build a solution to $\mathrm{DP}[j][a_1] \ldots [a_i] \ldots [a_g]$ by using the solution of $\mathrm{DP}[j - 1][a_1] \ldots [a_i] \ldots [a_g]$. In this case if the length of the LCS were to increase, then we note that the last element of the new LCS has to be $\pi[j]$ (otherwise, the longer LCS we find would also have been valid for $\mathrm{DP}[j - 1][a_1] \ldots [a_i] \ldots [a_g]$). So the solution to this problem can be obtained by appending $\sigma_{j-1}(a_1, \ldots, a_i - 1 \ldots a_g)$ (for the appropriate group $i \in [g]$) with $\pi[j]$ to get a LCS of length $1 + \mathrm{DP}[j - 1][a_1] \ldots [a_i - 1] \ldots [a_g]$. Hence we note that this case essentially reduces to case 1B (note that this case hence does not feature in the recurrence, and is just mentioned for completeness).

Therefore, by iterating over all groups to consider all possible candidates, we get the recurrence,

$$\mathrm{DP}[j][a_1] \ldots [a_g] = \max_{i \in [g]} \begin{cases} \mathrm{DP}[j][a_1] \ldots [a_i - 1] \ldots [a_g] + 1, \text{ if } \pi[1 \ldots j] \text{ has } \geq a_i \text{ elements of } G_i; \\ \mathrm{DP}[j][a_1] \ldots [a_i - 1] \ldots [a_g], \text{ if } \pi[1 \ldots j] \text{ has } \leq a_i - 1 \text{ elements of } G_i. \\ \mathrm{DP}[j - 1][a_1] \ldots [a_i - 1] \ldots [a_g] + 1, \text{ if } \pi[j] \text{ is in } G_i; \\ \mathrm{DP}[j - 1][a_1] \ldots [a_i - 1] \ldots [a_g], \text{ if } \pi[j] \text{ is not in } G_i; \end{cases}$$

Also note that here we use the fact that the length of the LCS can only increase by one in any of the above cases. If the LCS increased by more than one, we can ignore one of the characters (being an arbitrary character, or the newly considered character from the prefix, corresponding to the appropriate case) of the newly obtained LCS, from its corresponding solution string $\sigma$, thereby

obtaining a solution with a larger LCS for a previously solved subproblem. This would hence give us a contradiction.

To ensure that only subproblems where the top-$\ell$ ranks satisfy the fairness constraints ($\forall i \in [g], \lfloor \alpha_i \ell \rfloor \leq a_i \leq \lceil \beta_i \ell \rceil$ and $\forall i \in [g], a_i \leq |G_i|$) are used for construction, we set all 'invalid' subproblems, to have a value of $-\infty$ (we do this only for prefixes of size larger than $k$).

For the *base case*, we have that for any $j \in [d]$ and all groups $i \in [g]$, $\text{DP}[j][0] \ldots [a_i = 1] \ldots [0] = 1$, if there is a element of $G_i$ in $\pi[1 \ldots j]$; and $0$ otherwise. Note that the above recurrence can be solved in a top-down approach. For a formal description of the pseudocode of the algorithm, see Algorithm 3.

There are $\mathcal{O}(d^{g+1})$ DP subproblems, evaluating each of which takes $\mathcal{O}(d)$ time. Then, we iterate over all the possible valid fair strings $\sigma(a_1, \ldots, a_g)$ (there are at most $\mathcal{O}(d^g)$ such strings) to find the one with the longest possible common subsequence. This gives us an overall running time of $\mathcal{O}(d^{g+2})$. $\qquad\square$

## A.3   Fair Rank Aggregation: First Meta Algorithm

---

**Algorithm 4:** Meta-algorithm 1 for the $q$-mean fair rank aggregation.

---
**Input:** A set $S \subseteq \mathcal{S}_d$ of $n$ rankings.
**Output:** A $(c+2)$-approximate fair aggregate rank of $S$.
1 Initialize $S' \leftarrow \emptyset$
2 **for** *each point $\pi$ in $S$* **do**
3 $\quad$ Find a $c$-approximate closest fair ranking $\sigma$ to $\pi$ using the algorithm $\mathcal{A}$
4 $\quad$ $S' \leftarrow S' \cup \{\sigma\}$
5 Initialize $\bar{\sigma} \leftarrow \emptyset$
6 Initialize $\text{Obj}_q(S, \sigma) \leftarrow \infty$
7 **for** *each point $\sigma$ in $S'$* **do**
8 $\quad$ **if** $\text{Obj}_q(S, \sigma) < \text{Obj}_q(S, \bar{\sigma})$ **then**
9 $\quad\quad$ $\bar{\sigma} \leftarrow \sigma$
10 **return** $\bar{\sigma}$

---

**Lemma 4.4.** *Given a set $S \subseteq \mathcal{S}_d$ of $n$ rankings, let $\sigma^*$ be an optimal $q$-mean fair aggregated rank of $S$ under a distance function $\rho$. Further, let $\bar{\pi}$ be a nearest neighbor (closest ranking) of $\sigma^*$ in $S$, and $\bar{\sigma}$ be a $c$-approximate closest fair ranking to $\bar{\pi}$, for some $c \geq 1$. Then $\forall \pi \in S, \rho(\pi, \bar{\sigma}) \leq (c+2) \cdot \rho(\pi, \sigma^*)$.*

*Proof.* Since $\sigma^*$ is a fair ranking and $\bar{\sigma}$ is a $c$-approximate closest fair ranking to $\bar{\pi}$

$$\rho(\bar{\pi}, \bar{\sigma}) \leq c \cdot \rho(\bar{\pi}, \sigma^*). \tag{1}$$

Since $\bar{\pi} \in S$ is a closest ranking to $\sigma^*$ in the set $S$,

$$\forall \pi \in S, \rho(\bar{\pi}, \sigma^*) \leq \rho(\pi, \sigma^*). \tag{2}$$

Then it follows from Equation 1,

$$\forall \pi \in S, \rho(\bar{\pi}, \bar{\sigma}) \leq c \cdot \rho(\pi, \sigma^*) \tag{3}$$

Now, for any $\pi \in S$, we get,

$$\begin{aligned}
\rho(\pi, \bar{\sigma}) &\leq \rho(\pi, \sigma^*) + \rho(\sigma^*, \bar{\sigma}) && \text{(By the triangle inequality)} \\
&\leq \rho(\pi, \sigma^*) + \rho(\sigma^*, \bar{\pi}) + \rho(\bar{\pi}, \bar{\sigma}) && \text{(By the triangle inequality)} \\
&\leq \rho(\pi, \sigma^*) + \rho(\pi, \sigma^*) + c \cdot \rho(\pi, \sigma^*) && \text{(By Equation 2 and Equation 3)} \\
&\leq (c+2) \cdot \rho(\pi, \sigma^*). && \square
\end{aligned}$$

---

**Algorithm 5:** Alternate Meta-algorithm for finding fair-aggregate rank.

---

**Input:** A set $S \subseteq \mathcal{S}_d$ of $n$ rankings.

**Output:** A $(c_1 c_2 + c_1 + c_2)$-approximate fair aggregate rank of $S$.

1 Call $\mathcal{A}_1(S)$ to find the $c_1$-approximate aggregate rank $\pi^*$ of the input set $S$.

2 Call $\mathcal{A}_2(\pi^*)$ to find the $c_2$-approximate closest fair ranking $\bar{\sigma}$, to $\pi^*$.

3 **return** $\bar{\sigma}$.

---

## A.4 Fair Rank Aggregation: Second Meta Algorithm

**Lemma 4.9.** *Given a set $S \subseteq \mathcal{S}_d$ of $n$ rankings, let $\sigma^*$ be an optimal $q$-mean fair aggregated rank of $S$ under a distance function $\rho$. Further, let $\pi^*$ be the $c_1$-approximate aggregate rank of $S$ and $\bar{\sigma}$ be a $c_2$-approximate closest fair ranking to $\pi^*$, for some $c_1, c_1 \geq 1$. Then*

$$\forall \pi \in S, \rho(\pi, \bar{\sigma}) \leq (c_1 c_1 + c_1 + c_2) \cdot \rho(\pi, \sigma^*).$$

*Proof.* Since $\pi^*$ is a $c_1$-approximate rank aggregation of the input, we have that,

$$\rho(\pi, \pi^*) \leq c_1 \cdot \rho(\pi, \sigma^*). \tag{4}$$

Since $\bar{\sigma}$ is the $c_2$-approximate closest fair rank to $\pi^*$, we have,

$$\rho(\pi^*, \bar{\sigma}) \leq c_2 \cdot \rho(\pi^*, \sigma^*) \tag{5}$$

So, for any $\pi \in S$ we get,

$$
\begin{aligned}
\rho(\pi, \bar{\sigma}) &\leq \rho(\pi, \pi^*) + \rho(\pi^*, \bar{\sigma}) && \text{(By the triangle inequality)} \\
&\leq \rho(\pi, \pi^*) + c_2 \cdot \rho(\pi^*, \sigma^*) && \text{(By Equation 5)} \\
&\leq \rho(\pi, \pi^*) + c_2 \cdot (\rho(\pi^*, \pi) + \rho(\pi, \sigma^*)) && \text{(By the triangle inequality)} \\
&\leq (1 + c_2) \cdot \rho(\pi, \pi^*) + c_2 \cdot \rho(\pi, \sigma^*) && \\
&\leq (1 + c_2) \cdot c_1 \cdot \rho(\pi, \sigma^*) + c_2 \cdot \rho(\pi, \sigma^*) && \text{(By Equation 4)} \\
&\leq (c_1 c_2 + c_1 + c_2) \cdot \rho(\pi, \sigma^*). && \square
\end{aligned}
$$

**Corollary 4.10.** *For $q = 1$, there exists an $\mathcal{O}(d^3 \log d + n^2 d + nd^2)$ time meta-algorithm, that finds a 3-approximate solution to the $q$-mean fair rank aggregation problem (i.e., the fair median problem) under Spearman footrule metric.*

*Proof.* In [DKNS01] it is shown that optimal rank aggregation under Spearman footrule can be reduced in polynomial time to the minimum cost perfect matching in a bipartite graph. The reduction takes $\mathcal{O}(nd^2)$ time, which is needed to compute the edge weights for the constructed bipartite graph. Further, [vdBLN+20] gives a randomized $\tilde{\mathcal{O}}(m + n^{1.5})$ time algorithm for the minimum cost perfect bipartite matching problem for bipartite graphs with $n$ nodes and $m$ edges. The reduction in [DKNS01] creates an instance of the minimum cost perfect bipartite matching problem with $O(d)$ nodes and $O(d^2)$ edges. Hence, the result of [vdBLN+20] gives us an exact $\tilde{\mathcal{O}}(nd^2)$ time rank aggregation algorithm for Spearman footrule, i.e., $c_1 = 1$ and $t_1 = \tilde{\mathcal{O}}(n^2 + d)$.

Since we can compute an exact closest fair ranking under Spearman footrule in $\mathcal{O}(d^3 \log d)$, we have that $c_2 = 1$, and $t_2 = \mathcal{O}(d^3 \log d)$. Moreover, we can also compute the Spearman footrule distance in $f(d) = \mathcal{O}(d)$ time. Now plugging in these values to Theorem 4.8, we get that we can find a 3-approximate solution to the fair median problem under Spearman footrule in $\mathcal{O}(d^3 \log d + n^2 d + nd^2)$ time. $\square$

## B  Better approximation algorithm for Ulam fair median

In this section, we provide a polynomial time $(3 - \varepsilon)$-approximation algorithm for the fair median problem under the Ulam metric for constantly many groups (proving Theorem 4.11). More specifically, we show the following theorem.

**Theorem B.1.** *For $q = 1$, there exists a constant $\varepsilon > 0$ and an algorithm that, given a set $S \subseteq \mathcal{S}_d$ of $n$ rankings of $d$ candidates where these candidates are partitioned into $g \geq 1$ groups, finds a $(3 - \varepsilon)$-approximate solution to the $q$-mean fair rank aggregation problem (i.e., the fair median problem), under the Ulam metric in time $\mathcal{O}(nd^{g+2} + n^2 d \log d)$.*

In proving the above theorem, we first describe the algorithm, then provide the running time analysis, and finally analyze the approximation factor.

**Description of the algorithm.** We run two procedures, each of which outputs a fair ranking (candidate fair median), and then return the one (among these two candidates) with the smaller objective value. The first procedure is the one used in Corollary 4.7. The second procedure is based on the relative ordering of the candidates and is very similar to that used in [CDK21] with the only difference being that now we want the output ranking to be fair (which was not required in [CDK21]). Our second procedure Algorithm 6 is given a parameter $\alpha \in [0, 1/10]$. It first creates a directed graph $H$ with the vertex set $V(H) = [d]$ and the edge set $E(H) = \{(a, b) \mid a <_\pi b$ for at least $(1 - 2\alpha)n$ rankings $\pi \in S\}$. If this graph $H$ is acyclic, then we are done. Otherwise, we make $H$ acyclic as follows: Iterate over all the vertices $v \in V(H)$, and in each iteration, find a shortest cycle containing $v$, and then delete all the vertices (along with all the incident edges) in that cycle. In the end, we will be left with an acyclic subgraph $\bar{H}$ (of the initial graph $H$). Then, we perform a topological sorting to find an ordering $O$ of the vertex set $V(\bar{H})$. Note, $O$ need not be a ranking of $d$ candidates (since $V(\bar{H})$ could be a strict subset of $[d]$). Let $\bar{\sigma}_{par}$ be the sequence denoting the ordering $O$. Then we find a fair ranking $\bar{\sigma}$ that maximizes the length of an LCS with $\bar{\sigma}_{par}$ and output it.

---

**Algorithm 6:** Relative Order Algorithm

**Input:** A set $S \subseteq \mathcal{S}_d$ of $n$ rankings, $\alpha \in [0, 1/10]$.
**Output:** A fair ranking from $\mathcal{S}_d$.

1   $H \leftarrow ([d], E)$ where $E = \{(a, b) \mid a <_\pi b$ for at least $(1 - 2\alpha)n$ rankings $\pi \in S\}$
2   **for** *each point $v \in V(H) = [d]$* **do**
3      $C \leftarrow$ A shortest cycle containing $v$
4      $H = H - V(C)$
5   $\bar{H} \leftarrow H$
6   $\bar{\sigma}_{par} \leftarrow$ Sequence representing a topological ordering of $V(\bar{H})$
7   Find a fair ranking $\bar{\sigma} \in \mathcal{S}_d$ that maximizes the length of a LCS with $\bar{\sigma}_{par}$
8   **return** $\bar{\sigma}$

---

**Running time analysis.** The first procedure that is used in Corollary 4.7 takes $\mathcal{O}(nd^{g+2} + n^2 d \log d)$ time. It follows from [CDK21], Step 1-6 of the second procedure (Algorithm 6) takes time $\mathcal{O}(nd^2 + d^3) = \mathcal{O}(nd^{g+2})$ (since $g \geq 1$). To perform Step 7 of Algorithm 6, we use the dynamic programming described in Algorithm 3. So, this step takes $\mathcal{O}(d^{g+2})$ time. As finally, we output the fair ranking produced by the two procedures that have a smaller objective value; the total running time is $\mathcal{O}(nd^{g+2} + n^2 d \log d)$.

**Showing the approximation guarantee.** Although the analysis proceeds in a way similar to that in [CDK21], it differs significantly in many places since now the output of our algorithm must be a fair ranking (not an arbitrary one). Let $\sigma^*$ be an (arbitrary) fair median (1-mean fair aggregated rank) of $S$ under the Ulam metric. Then $\texttt{OPT}(S) = \sum_{\pi \in S} \mathcal{U}(\pi, \sigma^*)$, which for brevity we denote by $\texttt{OPT}$. For any $S' \subseteq S$, let $\texttt{OPT}_{S'} := \sum_{\pi \in S'} \mathcal{U}(\pi, \sigma^*)$. Let us take parameters $\alpha, \beta, \gamma, \varepsilon, \eta, \nu$, the value of which will be set later. Let us consider the following set of fair rankings

$$S_f := \{\sigma \in \mathcal{S}_d \mid \sigma \text{ is a fair ranking closest to } \pi \in S\}.$$

It is worth noting that our first procedure (the algorithm used in Corollary 4.7) essentially outputs a fair ranking from the set $S_f$ with the smallest objective value.

Assume that

$$\forall_{\sigma \in S_f}, \ \mathcal{U}(\sigma, \sigma^*) > (2 - \varepsilon)\texttt{OPT}/n. \tag{6}$$

Otherwise, let $\sigma' \in S_f$ does not satisfy the above assumption, i.e., $\mathcal{U}(\sigma', \sigma^*) \le (2 - \varepsilon)\mathtt{OPT}/n$. Then

$$\sum_{\pi \in S} \mathcal{U}(\pi, \sigma') \le \sum_{\pi \in S} (\mathcal{U}(\pi, \sigma^*) + \mathcal{U}(\sigma^*, \sigma')) \qquad \text{(By the triangle inequality)}$$
$$= \mathtt{OPT} + n \cdot \mathcal{U}(\sigma^*, \sigma')$$
$$\le (3 - \varepsilon)\mathtt{OPT}. \tag{7}$$

So from now on, we assume 6. It then directly follows from the triangle inequality that

$$\forall_{\pi \in S}, \ \mathcal{U}(\pi, \sigma^*) > (1 - \varepsilon/2)\mathtt{OPT}/n. \tag{8}$$

For each $\pi \in S \cup S_f$, consider an (arbitrary) LCS $\ell_\pi$ between $\pi$ and $\sigma^*$. Let $I_\pi$ denote the set of symbols in $[d]$ that are not included in $\ell_\pi$. Note, $|I_\pi| = d - |LCS(\pi, \sigma^*)| = \mathcal{U}(\pi, \sigma^*)$.

For any $\pi \in S \cup S_f$ and $D \subseteq [d]$, let $I_\pi(D) := I_\pi \cap D$. For any $a \in [d]$ and $S' \subseteq S$,

$$\mathtt{cost}_{S'}(a) := |\{\pi \in S' \mid a \text{ is not included in } \ell_\pi\}|.$$

When $S' = S$, for brevity, we drop the subscript $S'$. For any $D \subseteq [d]$ and $S' \subseteq S$, $\mathtt{OPT}_{S'}(D) := \sum_{a \in D} \mathtt{cost}_{S'}(a)$. When $D = [d]$, for brevity, we only use $\mathtt{OPT}_{S'}$. Note, $\mathtt{OPT} = \mathtt{OPT}_S$.

We call a symbol $a \in [d]$ *lazy* if $\mathtt{cost}(a) \le \alpha n$; otherwise *active*. Let $L$ denote the set of all lazy symbols, and $A = [d] \setminus L$ (i.e., the set of all active symbols).

**Case 1:** $|A| \le \beta \cdot \mathtt{OPT}/n$

We use the following result from [CDK21].

**Claim B.2** ([CDK21]). *The set of symbols in $L \cap V(\bar{H})$ forms a common subsequence between $\bar{\sigma}_{par}$ and $\sigma^*$. Furthermore, $|L \setminus V(\bar{H})| \le \frac{4\alpha}{1 - 4\alpha}|A|$.*

Using the above claim, we show the following lemma.

**Lemma B.3.** *Consider an $\alpha \in [0, 1/10]$ and $\beta \in (0, 1)$. Given an input set $S \subseteq \mathcal{S}_d$ of size $n$, let the set of active symbols $A$ be of size at most $\beta \cdot \mathtt{OPT}/n$. Then on input $S, \alpha$, Algorithm 6 outputs a fair ranking $\bar{\sigma}$ that is an $(1 + 3\beta(1 + 8\alpha))$-approximate fair median (1-mean fair aggregated rank) of $S$.*

*Proof.* First note, by Claim B.2, the set of symbols in $L \cap V(\bar{H})$ forms a common subsequence between $\bar{\sigma}_{par}$ and $\sigma^*$. Now since $\sigma^*$ is a fair ranking, the length of an LCS between $\bar{\sigma}_{par}$ and $\bar{\sigma}$ must be at least $|L \cap V(\bar{H})| - |V(\bar{H}) \setminus L|$. Moreover,

$$|LCS(\bar{\sigma}, \sigma^*)| \ge |L \cap V(\bar{H})| - 2|V(\bar{H}) \setminus L| \ge |L \cap V(\bar{H})| - 2|A|. \tag{9}$$

(Note, since a fair ranking $\sigma^*$ exists, it is always possible to perform Step 7 of Algorithm 6 to output a fair ranking $\bar{\sigma}$.)

$$\mathtt{Obj}(S, \bar{\sigma}) = \sum_{\pi \in S} \mathcal{U}(\pi, \bar{\sigma})$$
$$\le \sum_{\pi \in S} (\mathcal{U}(\pi, \sigma^*) + \mathcal{U}(\sigma^*, \bar{\sigma})) \qquad \text{(By the triangle inequality)}$$
$$= \mathtt{OPT} + n \cdot (d - |LCS(\sigma^*, \bar{\sigma})|)$$
$$\le \mathtt{OPT} + n \cdot (d - |L \cap V(\bar{H})| + 2|A|) \qquad \text{(By Equation 9)}$$
$$= \mathtt{OPT} + n \cdot (|A| + |L| - |L \cap V(\bar{H})| + 2|A|)$$
$$= \mathtt{OPT} + n \cdot (3|A| + |L \setminus V(\bar{H})|)$$
$$\le \mathtt{OPT} + n \cdot \frac{3}{1 - 4\alpha}|A| \qquad \text{(By Claim B.2)}$$
$$\le \mathtt{OPT} + \frac{3\beta}{1 - 4\alpha}\mathtt{OPT} \qquad \text{(Since } |A| \le \beta \cdot \mathtt{OPT}/n\text{)}$$
$$\le (1 + 3\beta(1 + 8\alpha))\mathtt{OPT} \qquad \text{(Since } \alpha \in [0, 1/10]\text{)}.$$

$\square$

**Case 2:** $|A| > \beta \cdot \text{OPT}/n$

Recall, we consider parameters $\alpha, \beta, \gamma, \varepsilon, \eta, \nu$, the value of which will be set later. Let us partition $S$ into the following sets of *near* and *far* rankings: Let $N := \{\pi \in S \mid \mathcal{U}(\pi, \sigma^*) < (1 + \varepsilon/\alpha)\text{OPT}/n\}$, and $F := S \setminus N$.

**Claim B.4** ([CDK21])**.**

$$OPT_N \geq (1 - \alpha/2)(1 - \varepsilon/2)OPT. \tag{10}$$

$$OPT_N(A) \geq \frac{\alpha n}{2}|A| \geq \frac{\alpha\beta}{2}OPT_N \tag{11}$$

Let $R := \{\pi \in N \mid |I_\pi(A)| \geq (1 - \gamma)\frac{\alpha}{2}|A|\}$. Then

$$\text{OPT}_{N \setminus R}(A) \leq (1 - \gamma)\frac{\alpha}{2}|A| \cdot |N \setminus R|$$
$$\leq (1 - \gamma)\text{OPT}_N(A) \qquad \text{(By Equation 11).} \tag{12}$$

As a consequence, we get that

$$\text{OPT}_R(A) \geq \gamma\text{OPT}_N(A). \tag{13}$$

Further partition $R$ into $R_1, \ldots, R_r$ for $r = \lceil \log_{1+\nu}(2/(\alpha - \alpha\gamma)) \rceil$ as follows

$$R_i := \{\pi \in R \mid (1 + \nu)^{i-1}(1 - \gamma)\frac{\alpha}{2}|A| < |I_\pi(A)| \leq (1 + \nu)^i(1 - \gamma)\frac{\alpha}{2}|A|\}. \tag{14}$$

By an averaging argument, there exists $i^* \in [r]$ such that for $R^* = R_{i^*}$,

$$\text{OPT}_{R^*}(A) \geq \text{OPT}_R(A)/r \geq \frac{\gamma}{r}\text{OPT}_N(A) \tag{15}$$

where the last inequality follows from Equation 13.

Let us consider $\eta = \frac{1}{2}\left((1 + \nu)^{i^*-1}(1 - \gamma)\alpha/2\right)^2$. For each $\pi \in S$, let $\sigma(\pi)$ be the closest fair ranking that is in the set $S_f$. Next, consider the following procedure to segregate $R^*$ into a set of clusters with $\mathcal{C}$ denoting the set of cluster centers. (We emphasize that the following clustering step is only for the sake of analysis.)

1. Initialize $\mathcal{C} \leftarrow \emptyset$.

2. Iterate over all $\pi \in R^*$ (in the non-decreasing order of $|I_\pi(L)|$)

   (a) If for all $\pi' \in \mathcal{C}$, $|I_{\sigma(\pi')} \cap I_\pi(A)| < \eta|A|$, then add $\pi$ in $\mathcal{C}$. Also, create a cluster $C_\pi \leftarrow \{\pi\}$.

   (b) Else pick some $\pi' \in \mathcal{C}$ (arbitrarily) such that $|I_{\sigma(\pi')} \cap I_\pi(A)| \geq \eta|A|$ and add $\pi$ in $C_{\pi'}$.

Since we process all $\pi \in R^*$ in the non-decreasing order of $|I_\pi(L)|$, it is straightforward to see

$$\forall_{\pi' \in \mathcal{C}}, \ \forall_{\pi \in C_{\pi'}}, \ |I_{\pi'}(L)| \leq |I_\pi(L)|. \tag{16}$$

Next, we use the following simple combinatorial lemma from [CDK21].

**Lemma B.5** ([CDK21])**.** *For any $c, d \in \mathbb{N}$ and $\eta \in (0, \frac{1}{2c^2}]$, any family $\mathcal{F}$ of subsets of $[d]$ where*

- *Every subset $I \in \mathcal{F}$ has size at least $n/c$, and*

- *For any two $I \neq J \in \mathcal{F}$, $|I \cap J| \leq \eta d$,*

*has size at most $2c$.*

We use the above lemma to prove the following.

**Claim B.6.** *For any $\alpha \in (0, 1)$ and $\gamma \in (0, 0.5)$, $|\mathcal{C}| \leq 8/\alpha$.*

*Proof.* Recall, by definition, for any $\pi' \in \mathcal{C}$, $\sigma(\pi') \in S_f$. By Assumption 6, for any $\sigma \in S_f$,

$$|I_\sigma| > (2 - \varepsilon)\mathtt{OPT}/n \geq (2 - \varepsilon)\alpha|A|$$

where the last inequality follows since by the definition of active symbols, $\mathtt{OPT} \geq \alpha n|A|$. Also, by Equation 14, for each $\pi \in R^*$, $|I_\pi(A)| \geq (1 + \nu)^{i^*-1}(1 - \gamma)\frac{\alpha}{2}|A|$. Then it directly follows from Lemma B.5, $|\mathcal{C}| \leq 2\left\lceil \frac{1}{(1+\nu)^{i^*-1}(1-\gamma)\frac{\alpha}{2}} \right\rceil \leq 8/\alpha$ (for $\gamma < 0.5$). □

Since $C_{\pi'}$'s create a partitioning of the set $R^*$, by averaging,

$$\exists_{\pi' \in \mathcal{C}}, \ \mathtt{OPT}_{C_{\pi'}}(A) \geq \frac{\mathtt{OPT}_{R^*}(A)}{|\mathcal{C}|}$$

$$\geq \frac{\alpha\gamma}{8r}\mathtt{OPT}_N(A) \qquad \text{(By Equation 15 and Claim B.6).} \qquad (17)$$

From now on, fix a $\tilde{\pi} \in \mathcal{C}$ that satisfies the above. Then,

$$\mathtt{OPT}_{C_{\tilde{\pi}}} \geq \mathtt{OPT}_{C_{\tilde{\pi}}}(A) \geq \frac{\alpha\gamma}{8r}\mathtt{OPT}_N(A) \qquad \text{(By Equation 17)}$$

$$\geq \frac{\alpha^2\beta\gamma}{16r}\mathtt{OPT}_N \qquad \text{(By Equation 11).} \qquad (18)$$

Note, $\mathtt{OPT}_{C_{\tilde{\pi}}}(L) + \mathtt{OPT}_{C_{\tilde{\pi}}}(A) = \mathtt{OPT}_{C_{\tilde{\pi}}} \leq \mathtt{OPT}_N$, and thus,

$$\mathtt{OPT}_{C_{\tilde{\pi}}}(L) \leq \left(1 - \frac{\alpha^2\beta\gamma}{16r}\right)\mathtt{OPT}_{C_{\tilde{\pi}}}. \qquad (19)$$

Let $\tilde{\sigma} = \sigma(\tilde{\pi})$. Observe, since $\mathcal{U}(\tilde{\pi}, \tilde{\sigma}) \leq \mathcal{U}(\tilde{\pi}, \sigma^*)$ (recall, $\sigma^*$ is a fair ranking and $\tilde{\sigma}$ is a closest fair ranking to $\tilde{\pi}$), by the triangle inequality,

$$\mathcal{U}(\sigma^*, \tilde{\sigma}) \leq 2\mathcal{U}(\sigma^*, \tilde{\pi}). \qquad (20)$$

Next, take any far ranking $\pi \in F$.

$$\mathcal{U}(\pi, \tilde{\sigma}) \leq \mathcal{U}(\pi, \sigma^*) + \mathcal{U}(\sigma^*, \tilde{\sigma}) \qquad \text{(By the triangle inequality)}$$
$$\leq \mathcal{U}(\pi, \sigma^*) + 2\mathcal{U}(\sigma^*, \tilde{\pi}) \qquad \text{(By Equation 20)}$$
$$\leq 3\mathcal{U}(\pi, \sigma^*) \qquad \text{(Since } \pi \in F \text{ and } \tilde{\pi} \in N\text{).}$$

Hence, we get that

$$\forall_{\pi \in F}, \ \mathcal{U}(\pi, \tilde{\sigma}) \leq 3\mathcal{U}(\pi, \sigma^*). \qquad (21)$$

Then consider any near ranking $\pi \in N$.

$$\mathcal{U}(\pi, \tilde{\sigma}) \leq \mathcal{U}(\pi, \sigma^*) + \mathcal{U}(\sigma^*, \tilde{\sigma}) \qquad \text{(By the triangle inequality)}$$
$$\leq \mathcal{U}(\pi, \sigma^*) + 2\mathcal{U}(\sigma^*, \tilde{\pi}) \qquad \text{(By Equation 20)}$$
$$\leq \mathcal{U}(\pi, \sigma^*) + 2(1 + \varepsilon/\alpha)\mathtt{OPT}/n \qquad \text{(Since } \tilde{\pi} \in N\text{)}$$
$$\leq \mathcal{U}(\pi, \sigma^*) + 2\frac{1 + \varepsilon/\alpha}{1 - \varepsilon/2}\mathcal{U}(\pi, \sigma^*) \qquad \text{(By Assumption 8)}$$
$$\leq \frac{3 + (4/\alpha - 1)\varepsilon/2}{1 - \varepsilon/2}\mathcal{U}(\pi, \sigma^*).$$

Hence, we get that

$$\forall_{\pi \in N}, \ \mathcal{U}(\pi, \tilde{\sigma}) \leq \frac{3 + (4/\alpha - 1)\varepsilon/2}{1 - \varepsilon/2}\mathcal{U}(\pi, \sigma^*). \qquad (22)$$

Lastly, consider any $\pi \in C_{\tilde{\pi}}$. Note,

$$\mathcal{U}(\pi, \tilde{\sigma}) \leq |I_\pi| + |I_{\tilde{\sigma}}| - |I_\pi \cap I_{\tilde{\sigma}}|$$
$$\leq |I_\pi| + |I_{\tilde{\sigma}}| - |I_\pi(A) \cap I_{\tilde{\sigma}}|$$
$$\leq |I_\pi| + |I_{\tilde{\sigma}}| - \eta|A| \qquad \text{(By the construction).}$$

Hence,

$$
\begin{aligned}
\sum_{\pi \in C_{\tilde{\pi}}} \mathcal{U}(\pi, \tilde{\sigma}) &\leq \sum_{\pi \in C_{\tilde{\pi}}} \left( |I_\pi| + |I_{\tilde{\sigma}}| - \eta |A| \right) \\
&\leq \sum_{\pi \in C_{\tilde{\pi}}} \left( |I_\pi| + 2|I_{\tilde{\pi}}| - \eta |A| \right) && \text{(By Equation 20)} \\
&\leq \sum_{\pi \in C_{\tilde{\pi}}} \left( |I_\pi(L)| + 2|I_{\tilde{\pi}}(L)| \right) + \sum_{\pi \in C_{\tilde{\pi}}} \left( |I_\pi(A)| + 2|I_{\tilde{\pi}}(A)| - \eta |A| \right) \\
&\leq 3 \sum_{\pi \in C_{\tilde{\pi}}} |I_\pi(L)| + \sum_{\pi \in C_{\tilde{\pi}}} \left( |I_\pi(A)| + 2|I_{\tilde{\pi}}(A)| - \eta |A| \right) && \text{(By Equation 16)} \\
&\leq 3 \sum_{\pi \in C_{\tilde{\pi}}} |I_\pi(L)| + \sum_{\pi \in C_{\tilde{\pi}}} \left( |I_\pi(A)| + 2(1+\nu)|I_\pi(A)| - \eta |A| \right) && \text{(Since } \pi, \tilde{\pi} \in R^*) \\
&\leq 3 \sum_{\pi \in C_{\tilde{\pi}}} |I_\pi| - \sum_{\pi \in C_{\tilde{\pi}}} \left( \eta |A| - 2\nu |I_\pi(A)| \right) \\
&\leq 3 \sum_{\pi \in C_{\tilde{\pi}}} |I_\pi| - \left( \frac{2\eta}{\alpha(1+\nu)^{i^*}(1-\gamma)} - 2\nu \right) \sum_{\pi \in C_{\tilde{\pi}}} |I_\pi(A)| && \text{(Since } \pi \in R^*) \\
&= 3 \sum_{\pi \in C_{\tilde{\pi}}} |I_\pi| - \left( \frac{(1+\nu)^{i^*-2}(1-\gamma)\alpha}{4} - 2\nu \right) \sum_{\pi \in C_{\tilde{\pi}}} |I_\pi(A)| && \left(\text{Recall, } \eta = \frac{((1+\nu)^{i^*-1}(1-\gamma)\alpha/2)^2}{2}\right) \\
&= 3 \sum_{\pi \in C_{\tilde{\pi}}} |I_\pi| - \rho \sum_{\pi \in C_{\tilde{\pi}}} |I_\pi(A)| && \left(\text{Let } \rho = \frac{(1+\nu)^{i^*-2}(1-\gamma)\alpha}{4} - 2\nu\right) \\
&\leq (3-\rho) \sum_{\pi \in C_{\tilde{\pi}}} |I_\pi| + \rho \sum_{\pi \in C_{\tilde{\pi}}} |I_\pi(G)| \\
&= (3-\rho)\mathtt{OPT}_{C_{\tilde{\pi}}} + \rho \mathtt{OPT}_{C_{\tilde{\pi}}}(G) && \text{(By the definition)} \\
&\leq \left( 3 - \frac{\rho \alpha^2 \beta \gamma}{16r} \right) \mathtt{OPT}_{C_{\tilde{\pi}}} && \text{(By Equation 19).}
\end{aligned}
$$

$$(23)$$

Finally, we deduce that

$$
\begin{aligned}
\sum_{\pi \in S} \mathcal{U}(\pi, \tilde{\sigma}) &= \sum_{\pi \in F} \mathcal{U}(\pi, \tilde{\sigma}) + \sum_{\pi \in N \setminus C_{\tilde{\pi}}} \mathcal{U}(\pi, \tilde{\sigma}) + \sum_{\pi \in C_{\tilde{\pi}}} \mathcal{U}(\pi, \tilde{\sigma}) \\
&\leq 3\mathtt{OPT}_F + \frac{3 + (4/\alpha - 1)\varepsilon/2}{1 - \varepsilon/2} \mathtt{OPT}_{N \setminus C_{\tilde{\pi}}} + \left( 3 - \frac{\rho \alpha^2 \beta \gamma}{16r} \right) \mathtt{OPT}_{C_{\tilde{\pi}}} && \text{(By Equations 21, 22, 23)} \\
&\leq 3\mathtt{OPT} + (3 + 2/\alpha + (4/\alpha - 1)\varepsilon/2)\varepsilon \mathtt{OPT}_{N \setminus C_{\tilde{\pi}}} - \frac{\rho \alpha^2 \beta \gamma}{16r} \mathtt{OPT}_{C_{\tilde{\pi}}} && \text{(Assuming } \varepsilon < 0.5) \\
&\leq 3\mathtt{OPT} + (3 + 2/\alpha + (4/\alpha - 1)\varepsilon/2)\varepsilon \mathtt{OPT}_N - \frac{\rho \alpha^2 \beta \gamma}{16r} \mathtt{OPT}_{C_{\tilde{\pi}}} \\
&\leq 3\mathtt{OPT} - \left( \frac{\rho \alpha^4 \beta^2 \gamma^2}{256r^2} - (3 + 2/\alpha + (4/\alpha - 1)\varepsilon/2)\varepsilon \right) \mathtt{OPT}_N && \text{(By Equation 18)} \\
&\leq \left( 3 - (1 - \alpha/2)(1 - \varepsilon/2)(\frac{\rho \alpha^4 \beta^2 \gamma^2}{256r^2} - (3 + 2/\alpha + (4/\alpha - 1)\varepsilon/2)\varepsilon) \right) \mathtt{OPT} && \text{(By Equation 10).}
\end{aligned}
$$

Consider an $\alpha \in (0, 1/10]$ and $\beta \in (0, 1)$. Then set $\gamma = 1/4$, $\nu = (1-\gamma)\alpha/32$. Now, it is not hard to verify that the constant $(1 - \alpha/2)(1 - \varepsilon/2)(\frac{\rho \alpha^4 \beta^2 \gamma^2}{256r^2} - (3 + 2/\alpha + (4/\alpha - 1)\varepsilon/2)\varepsilon) > 0$ for a proper choice of $\varepsilon < 0.5$ (which is a function of $\alpha, \beta$). Hence, we conclude the following.

**Lemma B.7.** *Consider an $\alpha \in (0, 1/10]$ and $\beta \in (0, 1)$. There exists a constant $\delta > 0$ such that given an input set $S \subseteq \mathcal{S}_d$ of size $n$, for which the set of active symbols $A$ is of size at least $\beta \cdot \mathtt{OPT}/n$, there exists a fair ranking $\tilde{\sigma} \in S_f$ such that $\mathtt{Obj}_1(S, \tilde{\sigma}) \leq (3 - \delta)\mathtt{OPT}$.*

Recall our first procedure (the algorithm used in Corollary 4.7) essentially outputs a fair ranking from the set $S_f$ with the smallest objective value. Let us set $\alpha = 1/10$ and $\beta = 1/6$. Now, the approximation guarantee of Theorem B.1 follows from Lemma B.3 and Lemma B.7.