# OpenReview forum: "Fair Rank Aggregation"
_NeurIPS.cc/2022/Conference — NeurIPS 2022 Accept_

### Official Review · Reviewer_YYph · 2022-07-11

**Rating:** 6
**Confidence:** 5
**Soundness:** 3 good
**Presentation:** 2 fair
**Contribution:** 2 fair

**Summary:**

This paper focused on the rank aggregation problem under a very general notion of proportional fairness. Authors propose a linear time exact algorithm to find the closest fair ranking (CFR) by a greedy strategy for Kendall tau and Ulam metrics and propose a novel algorithmic toolbox to solve a wide variety of rank aggregation objectives satisfying such generic fairness constraints. They provide rigorous mathematical derivation to prove that the there exists a linear time ranking algorithm.

**Questions:**

* This paper discusses the different cases by leveraging different distance function, Kendall tau and Ulam. But there is no comparison between these two different distance functions. Which one is more suitable for the fair rank aggregation task?

*This paper provides some mathematical derivation to compute the time complexity of the proposed algorithms. How about the performance of ranking by the proposed algorithms? Can you deliver some experiments to show the performance of your ranking, such as precision?


**Limitations:**

*This paper focuses on the mathematical derivation for the fair rank aggregation and illustrates that the proposed algorithms are better than concurrent work [WISR22] in line 132. Maybe authors can add more comparisons and give more detailed explanation for proving the superiority of the proposed algorithms.

*This paper studies group fairness, as well as proportional fairness. The assumption of this problem seems to be that each candidate only belongs to one group. I wonder if the algorithms in this paper still works if each candidate can belong to multiple groups at the same time.


**Strengths And Weaknesses:**

Strengths:
* This paper gives a novel meta-algorithm for the general rank aggregation problem under the fairness framework, which works for any generalized mean objective and any fairness criteria.
* This paper proves that the there exists a linear time ranking algorithm through rigorous mathematical derivation.
Weaknesses:
* The structure of this paper is incomplete, which makes the paper poor readable. This paper misses Sections of Related Works and Conclusion. Authors can refer to previous paper published in NeurIPS for revision.
* In this paper, authors only use one paragraph in Section 1.1 to introduce the comparison with only one concurrent work. It is difficult to convince me that the algorithm proposed in this paper is superior to the state-of-the-art algorithms.
* Some formulations are not correct. For example, in Section 1, the generalized mean objective in Section 1.1 $\sum_{i=1}^n (\rho(\pi_i, \sigma)^q)^{1/q}$ should be $(\sum_{i=1}^n \rho(\pi_i, \sigma)^q)^{1/q}$ in line 105.
* I think the contribution of the proposed algorithm is that authors design a novel objection function of greedy strategy for fair ranking and prove the algorithm is a linear time ranking algorithm. However, the greedy strategy for ranking is lack of innovation

---

> ### Author Response · Authors · 2022-08-02
> **Novelty of greedy, connections to other problems, and Generalizations**
>
> We thank the reviewer for the helpful comments.
>
> ---
>
> Comment: “This paper misses Sections of Related Works and Conclusion.”
>
> Response:  As suggested by the reviewers, we have now added more related works and a conclusion section in the revised version (while moving some of the proofs to the appendix). Due to space constraints, we initially only referred to the existing vast literature on ranking, rank aggregation, fair ranking, and fair rank aggregation. However, there are more problems that are also closely related, and we shall elaborate on such relations more in the final version of the paper.
>
> ---
>
> Comment: “Maybe authors can add more comparisons and give more detailed explanations for proving the superiority of the proposed algorithms.”
>
> Response:  We do not just claim that our algorithm is superior but rather that our results subsume the results from the previous work in [WISR22].
>
> [WISR22] considers the fair ranking problem under a setting that is closely related to ours. However, the fairness criterion in their work is much more restrictive. Their algorithms for CFR are only designed for a special case of our formulation where for each group $G_i$ and any position k in the output ranking, α_i = β_i = p(i), where $p(i)$ denotes the proportion of group $G_i$ in the entire population. Further, under the Kendall tau metric, they give a polynomial time exact algorithm for CFR only for the special case of binary groups (g = 2). They also provide additional algorithms for multiple groups - an exact algorithm that works in time exponential in the number of groups and a polynomial time 2-approximation. In contrast, we fully resolve the CFR problem under the Kendall tau metric by providing a linear time algorithm for the case of multiple groups and
> any arbitrary bounds on α_i and β_i for each group $G_i$. Further, we give the first results for CFR under the Ulam metric and FRA results for much more general q-mean objective functions (which incidentally gives the same algorithm as the one in [WISR22] when studied for the 1-mean objective under the Kendall tau metric). Also of note is that the formulation in [WISR22] does not allow for overlap between protected groups, whereas some of our algorithms do allow for such overlap (e.g., CFR under Ulam).
>
> ---
>
> Comment: “the greedy strategy for ranking is lack of innovation.”
>
> Response: We would like to refer to our rebuttal to Reviewer 2yve which also answers this comment in detail.
>
> ---
>
> Comment: “This paper discusses the different cases by leveraging different distance function, Kendall tau and Ulam. But there is no comparison between these two different distance functions. Which one is more suitable for the fair rank aggregation task?”
>
> Response: Kendall Tau and Ulam are two fundamental distance measures defined over permutations/rankings having two different sets of applications. Kendall-tau is perhaps the most commonly used one in social choice theory, web search, etc. (see, e.g. [DKNS01]). The importance of Kendall Tau lies in the fact that it is the only known measure to simultaneously satisfy neutrality, consistency, and the extended Condorcet property [Kem59, You 88]. Due to this, we have focused more on providing an efficient and simple algorithm for Kendall Tau in our paper.
>
> On the other hand, the Ulam metric has a wide range of applications in different consensus problems related to computational biology (e.g., see [CMS01] or [FLRTV09] mentioned below). Rank aggregation under Ulam has already been studied in [CDK21, CGJ21].
>
> However, we would like to remark that our reduction of fair rank aggregation to the problem of finding the closest fair ranking works under any metric.
>
> [FLRTV09] G. Fertin, A. Labarre, I. Rusu, É. Tannier, S. Vialette, Combinatorics of Genome Rearrangements, MIT Press, 2009.
>
> ---
>
> Comment: “The assumption of this problem seems to be that each candidate only belongs to one group. I wonder if the algorithms in this paper still works if each candidate can belong to multiple groups”
>
> Response: We would like to refer to our rebuttal to Reviewer 3bHj which also answers this comment in detail.
>
> ---
>
> Comment: “In this paper, authors only use one paragraph in Section 1.1 to introduce the comparison with only one concurrent work. It is difficult to convince me that the algorithm proposed in this paper is superior to the state-of-the-art algorithms.”
>
> Response: Fair rank aggregation is a new problem that has only recently become a subject of study. Hence, only a limited amount of literature is available for the same. But we would like to refer to our rebuttal to Reviewer VkxL, for a detailed comparison with some of the other recent works, where we argue that our algorithms are faster and apply to a more general setting.

---

> ### Comment · Reviewer_YYph · 2022-08-09
> **Reply**
>
> The author's response addressed my concerns.

---

### Official Review · Reviewer_2yve · 2022-07-11

**Rating:** 6
**Confidence:** 3
**Soundness:** 3 good
**Presentation:** 3 good
**Contribution:** 3 good

**Summary:**

The authors study fair rank aggregation, where candidates may belong to different groups and each group must be represented fairly in the final ranking (in a way that the designer can specify). They provide a linear time algorithm for finding the closest fair ranking under proportional fairness for the Kendall tau distance metric, and a poly-time dynamic programming algorithm for the Ulam metric for a constant number of groups (which is the case in practice). They then study the fair rank aggregation problem, and show that many biased rankings can be aggregated into a fair ranking with only a small loss in quality for a variety of distance metrics.


**Questions:**

1. Have you considered fair rank aggregation algorithms that do not use closest fair ranking as a subroutine?


**Limitations:**

Yes

**Strengths And Weaknesses:**

Strengths:
+ The definition of proportional fairness is general and flexible, and nicely generalizes previous results.
+ The fair rank aggregation problem is also an interesting line of inquiry, where the goal is to find a fair ranking that minimizes the generalized mean distance to the input profile.

Strength/weakness:
+ The meta-algorithms for fair rank aggregation are also simple (which could be a weakness, but as a first foray into the field seems like a plus), where the ideas are either to (1) find fair approximations of each input ranking and then choose the most central one, or (2) find a fair approximation of the (unfair) ranking that minimizes the generalized mean objective. However, there is perhaps more room for more sophisticated algorithms that don't involve the CFR subproblem.

Weaknesses:
- The algorithms described in the paper are all relatively simple (even if the analysis is nontrivial). This is only a minor critique, however: as an early paper in this area, that is to be expected.

---

> ### Author Response · Authors · 2022-08-02
> **Non-triviality of the greedy algorithm**
>
> We thank the reviewer for the helpful comments.
>
> ---
>
> Comment: “The algorithms described in the paper are all relatively simple (even if the analysis is nontrivial). “
>
> Response: We believe that the algorithms being simple is a strength of our results since it makes the algorithm usable in practice. However, some of our analyses are fairly complex. E.g., the correctness of our greedy algorithm that solves the closest fair ranking (CFR) problem under the Kendall-tau metric (proof of Theorem 3.4) requires quite delicate exchange arguments to establish that our algorithm indeed gives the optimal solution. Although the greedy algorithm seems intuitive for the CFR problem at first glance, it is not. Previous work [WISR22] has relied on a polytime 2-approximation algorithm (using a reduction to matching) and an exponential time exact algorithm. In fact, [WISR22] hinted that this problem could be NP-hard by noting that the CFR essentially finds a perfect matching in a convex bipartite graph while minimizing the number of crossings and minimizing crossing for the general bipartite graph is already NP-hard. Our greedy algorithm essentially shows that the bipartite graphs arising in CFR are much simpler than the general ones in terms of finding minimum crossing perfect matching.
>
> The proof of Theorem 4.11 (Appendix B, pages 20-26 of the initially submitted version, pages 21-27 of the revised version) that shows (3-epsilon)-approximation for the fair rank aggregation (FRA) problem under Ulam is also quite intricate and exploits the extremal structure of the set of unaligned symbols with respect to an unknown optimal aggregated fair ranking. Breaking the 3-factor (attained by our meta-algorithm) is essentially highly challenging, and we have achieved that for Ulam, which is one of the significant contributions of this work. It is worth mentioning that for the standard rank aggregation (without fairness criterion), 2-approximation is straightforward, and breaking that bound was among one of the fundamental challenges (e.g., $(2-\epsilon)$-factor under Ulam achieved in [CDK21] that appeared in SODA’21, 4/3-factor under Kendall-tau in [ACN08] that initially appeared in STOC’05, PTAS for Kendall-tau [KMS07] that appeared in STOC’07). Breaking the 2-factor for the rank aggregation does not imply breaking the 3-factor for the FRA.
>
> ---
>
> Comment: “Have you considered fair rank aggregation algorithms that do not use closest fair ranking as a subroutine?”
>
> Response: We attempted other methods to get fair aggregation but could not get better theoretical guarantees. There are technical hurdles to maintaining the required properties since even if we are given an input set consisting of only fair rankings, there is no guarantee that the output of an arbitrary aggregation algorithm would be fair (or even a permutation).
> It is also interesting to note that the CFR is a special case of the FRA (when the input has only a single ranking); hence, by solving FRA, we are getting an algorithm for the CFR. So in some sense, it is impossible to avoid CFR for this problem altogether.

---

> > ### Comment · Reviewer_2yve · 2022-08-09
> > **Author response**
> >
> > Thank you for your response! I agree that simplicity of the algorithms can be a plus; it was only a minor critique. My overall rating has not changed, and I'd still argue for acceptance.

---

### Official Review · Reviewer_VkxL · 2022-07-12

**Rating:** 5
**Confidence:** 3
**Soundness:** 3 good
**Presentation:** 2 fair
**Contribution:** 2 fair

**Summary:**

From a fairness or diversity perspective, this paper examines ranking problems in which candidates belonging to different groups have a fair representation in the final rankings. Using a linear time exact algorithm, the algorithm will find the closest fair ranking for Kendall tau metric under strong fairness (where the final ranking is fair for all values of k) given the parameters for defining fair representation (number of candidates from a particular group in the top-k positions of the ranking).
The authors also provide an exact algorithm for finding the closest fair ranking for the Ulam metric under strong fairness when there are a few number of groups. Additionally, the authors propose a novel meta-algorithm for the general rank aggregation problem under the fairness framework.
In a nutshell, the main contribution is to develop a novel algorithmic toolbox for the fair rank aggregation that solves a variety of rank aggregation objectives satisfying such generic fairness constraints. An essential takeaway of this work is that a set of potentially biased rankings can be aggregated into a fair ranking with only a small loss in the quality of the ranking.

**Questions:**

See above. Have you implemented your algorithm to experiment on a real-world data?

**Limitations:**

I don't if they have discussed the limitations of their approach.

**Strengths And Weaknesses:**

The paper throughly studies the fair rank aggregation problem under different fairness constraints. It provides two meta-algorithms to approximate the fair aggregated ranking. There are a lot of Theorems, Claims, Lemmas and Definitions in the paper which is good in a sense because it adds details but overall makes the paper not easy to follow. I suggest moving some of them to appendix. One of the main sections of the paper (Fair Rank Aggregation) starts in page 7! While the paper looks interesting and novel, it is pure theoretical paper which I'm not sure is a good choice for this venue. The paper introduces a generic approach and argues it is a novel algorithm but it does not test this method on a simulated or real-world data. It would be interesting to see some experiments and comparisons with the methods discussed in [1] or [2] as these papers mentioned in the paper as closest approaches and they provide experiments to show the effectiveness of their method.

Missing related work: In line 43, Besides mentioned papers, [1] is also a new related work that outputs a fair ranking robust to label noise that maximizes the ranking utility subject to group fairness constraints based on exposure such as demographic parity. It is an in-processing method which presents a preferable trade-off between fairness and utility.

[1] Caitlin Kuhlman and Elke Rundensteiner. Rank aggregation algorithms for fair con- sensus. Proceedings of the VLDB Endowment, 13(12), 2020.

[2] David Wei, Md Moinul Islam, Baruch Schieber, and Senjuti Basu Roy. Rank aggrega- tion with proportional fairness. In SIGMOD, page To appear, 2022.

[3] Memarrast, Omid, Ashkan Rezaei, Rizal Fathony, and Brian Ziebart. "Fairness for Robust Learning to Rank." arXiv preprint arXiv:2112.06288 (2021).

---

> ### Author Response · Authors · 2022-08-02
> **Establishing a theoretical framework for the problem, and comparison with related works**
>
> We thank the reviewer for the helpful comments.
>
> ---
>
> Comment: “Have you implemented your algorithm to experiment on a real-world data?”
>
> Response: We have not implemented our algorithms as part of the current project. In this work, our primary focus was to fill the enormous gap between theoretical understanding and empirical solutions to this problem. We would like to emphasize that the prior results (e.g., [KR20], also, another related, though not the same, variant in the below-mentioned reference [SNPS20]), with the sole exception of [WISR22], \emph{do not provide any provable guarantees} on the performance of the algorithms and are only empirically evaluated. Our paper is one of the initial papers that lays the theoretical framework required for studying fair rank aggregation. Please see the response to the following comment for a more detailed comparison with the previous works.
>
> [SNPS20] Maria Stratigi, Jyrki Nummenmaa, Evaggelia Pitoura, and Kostas Stefanidis. Fair sequential group recommendations. In SAC '20: The 35th {ACM/SIGAPP} Symposium on Applied Computing. 2020
>
> ---
>
> Comment: “It would be interesting to see some experiments and comparisons with the methods discussed in [1] or [2] as these papers mentioned in the paper as closest approaches and they provide experiments to show the effectiveness of their method. ..In line 43, Besides mentioned papers, [3] is also a new related work .”
> Response: We first note that the notion of fairness that we consider is more general than that studied in all of the three papers. While our formulation allows for arbitrary upper/lower bounds (α_i, β_i resp. for a group $G_i$), all the three mentioned papers work with a special case of our formulation where, α_i = β_i (though with slightly different formulations in each case).
> Also, in the above (previous) works, it is assumed that there is no overlap between protected groups.
>
> To specifically address other differences and efficiency:
>
> In [1] the problem is solved using the heavy machinery of ILPs (or heuristic approaches). These methods are not very efficient as even in the paper they mention that the ILP techniques are not able to handle instances with a high number of candidates (of the order of d=60).
>
> [2] gives a polynomial time exact algorithm for CFR only for the special case of binary groups ($g = 2$). They also give additional algorithms for multiple groups - an exact algorithm that works in time exponential in the number of groups and a polynomial time 2-approximation. In contrast, we fully resolve the CFR problem under the Kendall tau metric by giving an (easy to implement) greedy algorithm for the case of multiple groups.
> Further our FRA meta-algorithms are for more general q-mean objective functions, and when studied for the 1-mean objective under Kendall tau metric, give the same algorithm as [2].
>
> Our paper studies a slightly different problem from the one considered in [3]. The aim of their problem is to output a matrix of probabilities $P_{n\times n}$, where $P_{ij}$ represents the probability of ranking an element i at rank j. It is not clear how one can obtain a valid ranking from this output. On the other hand we give a concrete permutation/ranking that satisfies the required constraints.
>
> In comparison to the above, we give combinatorial algorithms which are efficient, while simultaneously being simple (although some of the analyses are quite intricate).

---

### Official Review · Reviewer_3bHj · 2022-07-14

**Rating:** 4
**Confidence:** 4
**Soundness:** 3 good
**Presentation:** 3 good
**Contribution:** 2 fair

**Summary:**

The authors propose algorithms for closest fair ranking and fair rank aggregation problems.  Specifically, they consider two notions of fairness - weak : (a, b)-fair up to a prefix k of the ranking and strong : fair for all prefixes.  They provide an exact algorithm for the CFR problem for two metrics - Kendall Tau and Ulam.  Their results hold when the number of groups are a constant in the case of the Ulam metric.

**Questions:**

Do the algorithms extend to multi-dimensional spaces when a fairness criterion for each group can be represented as a d-dimensional vector.  The current algorithms are fairly straightforward because of the assumption of no 'overlap' between different groups.

**Limitations:**

While the problem of fair ranking is an important problem to consider, fair rank aggregation is not well motivated.

**Strengths And Weaknesses:**

Strengths:
1. Fair ranking is an important problem.
2. All the details are presented.  The paper is however a bit hard to read in some places.
3. The algorithms are fairly intuitive.

Weaknesses:
1. The algorithms are fairly simple to analyze.
2.  While fair ranking is an important problem, the same might not be the case for rank aggregation - one of the primary goals for aggregation is to overcome biases from any individual ranking.
3. Comparison to related work especially in fair group formation and fair clustering/partitioning is missing.  The latter problems are especially related under the weak notion of fairness when k is set to the size of each parition/cluster/group.

---

> ### Author Response · Authors · 2022-08-02
> **Motivation for FRA, connections to other problems, and Generalizations**
>
> We thank the reviewer for the helpful suggestions and comments.
>
> ---
>
> Comment: “While fair ranking is an important problem, the same might not be the case for rank aggregation - one of the primary goals for aggregation is to overcome biases from any individual ranking.”
>
> Response: Rank aggregation is a fundamental problem in multiple areas of computer science, including social choice theory, data mining, information retrieval, web search, etc.
> See the references mentioned in lines 30-33 in the paper for the importance of the rank aggregation problem. The fair rank aggregation has recently received significant attention [KR20, WISR22] to become a problem of prominence to the community. The reviewer correctly points out that rank aggregation, in principle, aims to remove bias from individual ranking. However, computing an aggregate ranking might potentially fail to achieve essential fairness goals like minority protection and restricted dominance. For instance, in one of the applications pointed out by reviewer 3bHj, fair committee formation: one would need to look at many votes and decide on the ranking (and hence the formed committee) that best represents all votes. However, the committee formed by just aggregating those votes (or rankings) may not well represent all the groups, especially the minority ones, an important goal to aspire for. To ensure that every group is fairly represented in the formed committee, it is required to incorporate that constraint while aggregating the votes.
>
> ---
>
> Comment: “Comparison to related work especially in fair group formation and fair clustering/partitioning is missing. The latter problems are especially related under the weak notion of fairness when k is set to the size of each partition/cluster/group.”
>
> Response: As suggested by the reviewers, we have now added more related works and a conclusion section in the revised version (while moving some of the proofs to the appendix). Due to space constraints, initially we only referred to the existing vast literature on ranking, rank aggregation, fair ranking, and fair rank aggregation. Indeed problems like clustering are also closely related (for instance fair rank aggregation can be considered as the fair 1-clustering problem where the input set is a set of rankings).
>
> ---
>
> Comment: “Do the algorithms extend to multi-dimensional spaces when a fairness criterion for each group can be represented as a d-dimensional vector. The current algorithms are fairly straightforward because of the assumption of no 'overlap' between different groups.”
>
> Response: We did consider generalizations of the problem and noted that our DP algorithm for CFR under Ulam works even for overlapping groups as long as the number of groups is constant. Further, the FRA meta-algorithms also work for general notions of fairness (with overlapping). So any (future) progress/improvement in the corresponding CFR problem would lead to an (approximate) solution to the FRA problem (even in case of the generalizations suggested by the reviewer).
>
> It is also worth mentioning that arbitrary CFR instances where a candidate belongs to more than two groups are known to be NP-hard [CSV18]. In fact, it is also known that these instances are hard to approximate to a factor better than $O{\Delta / \log\Delta}$ (where $\Delta$ is the maximum number of groups that any element belongs to), and hence it is unlikely to solve this problem without relaxing the notion of fairness. We note here that the multi-dimensional variant mentioned by the reviewer also captures this notion of overlapping groups and hence is also an NP-hard problem.
>
> ---
>
> Comment: “The algorithms are fairly simple to analyze.”
>
> Response: We would like to refer to our rebuttal to Reviewer 2yve which also answers this comment in detail.

---

### Meta-Review · Area_Chair_suCA · 2022-08-26

**Recommendation:** Accept
**Confidence:** Less certain

**Metareview:**

Reviewers liked the novelty in the fair range aggregation problem and enjoyed the simplicity of the proposed algorithms. The theoretical results are solid though the proof techniques are believed to be standard. There is a large room to improve the quality of presentation. Reviewers' opinions stayed the same after rebuttal and no additional points were raised during the discussion and no reviewer had a strong opinion on acceptance or rejection. So the paper  is on the slightly positive side of the fence.

**Award:**

No

---

### Decision · Program_Chairs · 2022-09-14

Accept